# SimLayerKV: A Simple Framework for Layer-Level KV Cache Reduction

## Abstract

Recent advancements in large language models (LLMs) have extended their capabilities to handle long contexts. However, increasing the number of model layers and the length of input sequences significantly escalates the memory required to store key-value (KV) cache, posing challenges for efficient inference. To mitigate this issue, we present SimLayerKV, a simple yet effective method that reduces inter-layer KV cache redundancies by selectively dropping cache in identified lazy layers. Our approach is based on the observation that certain layers in long-context LLMs exhibit "lazy" behavior, contributing less to modeling long-range dependencies compared to non-lazy layers. By analyzing attention weight patterns, we find that the behavior of these lazy layers is consistent across tokens during generation for a given input. This insight motivates our SimLayerKV, which identifies lazy layers and reduces their KV cache accordingly. SimLayerKV is training-free, generalizable, and can be implemented with only seven lines of code. We conduct extensive experiments on three representative LLMs, e.g., LLaMA2-7B, LLaMA3-8B, and Mistral-7B across 16 tasks from the LongBench benchmark. The results demonstrate that SimLayerKV achieves a KV cache compression ratio of 5× with only a 1.2% performance drop when combined with 4-bit quantization.

## 1 Introduction

Transformer-based autoregressive large language models (LLMs) have demonstrated exceptional performance across a wide range of tasks, such as question answering and arithmetic reasoning (Wei et al., 2022; Wang et al., 2022; Zhou et al., 2022; Yao et al., 2023). Recent advancements have extended their capabilities to handle long contexts, with models like Llama-3.1 supporting context lengths up to 128K tokens (Dubey et al., 2024) and Gemini-Pro-1.5 handling up to 1 million tokens (Reid et al., 2024). A critical component of these models during inference is the key-value (KV) cache, which stores precomputed key and value tensors for each token in the language sequence to avoid recomputing them for each attention layer. However, as the number of model layers and input lengths increases, the memory required for storing the KV cache grows significantly, posing challenges for inference efficiency (Zhang et al., 2024b; Wang et al., 2024a; Li et al., 2024). For example, with an input sequence length of 128K tokens, the memory required for the KV cache in Llama2-7B amounts to approximately 62.5 GB GPU memory, which is significantly larger than the 13.2 GB needed for the model parameters.

To address the challenge, various methods have recently been introduced to reduce the KV cache storage (Zhang et al., 2024b; Li et al., 2024; Hooper et al., 2024; Dong et al., 2024a; Yang et al., 2024c). One approach is quantization (Hooper et al., 2024; Dong et al., 2024a; Yang et al., 2024c; Dong et al., 2024b; Kang et al., 2024; Liu et al., 2024c; Sheng et al., 2023), which stores the KV cache in low-bit formats. Another approach resorts to eviction (Zhang et al., 2024b; Li et al., 2024; Zhang et al., 2024a; Yang et al., 2024b), which only preserves the most important tokens selected based on carefully crafted metrics. However, these works predominantly address intra-layer redundancies, neglecting the potential savings from inter-layer redundancies (Liu et al., 2024a), as illustrated in Figure 1.

Recent studies (Rajput et al., 2024; Brandon et al., 2024; Wu & Tu, 2024; Liao & Vargas, 2024; Wu & Tu, 2024; Liu et al., 2024a) have begun to explore inter-layer KV cache condense, leveraging redundancies across layers to reduce KV cache at the layer level. For example, Cross-Layer Attention

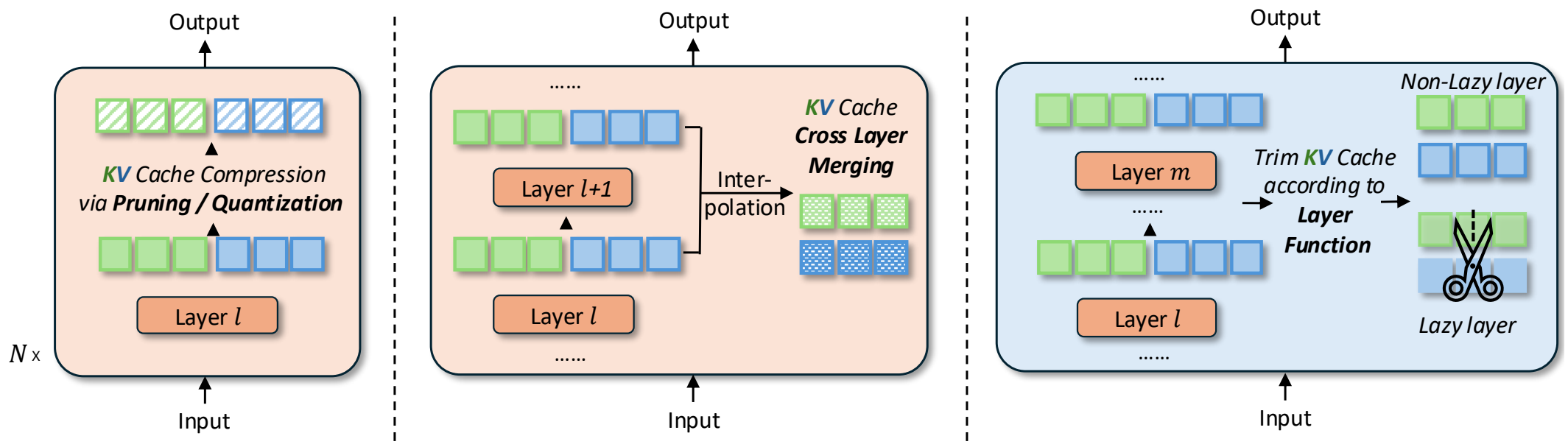

(a) Intra-layer methods (b) Inter-layer method: MiniCache (c) Inter-layer method: SimLayerKV

Figure 1: Comparison of intra-layer techniques (e.g., pruning and quantization) with two inter-layer methods: MinCache and our proposed SimLayerKV. (a) Intra-layer methods target KV redundancy within individual layers, applying compression independently to each layer; (b) MinCache reduces KV cache by merging adjacent layers through interpolation; (c) Our SimLayerKV selectively trims KV cache by identifying the functional role of each layer, reducing cache only in lazy layers.

(CLA) (Brandon et al., 2024) reuses the KV cache from the $n$-th layer for the subsequent $n$+1-th layer. While these methods are effective, they require additional training on existing LLMs (Rajput et al., 2024; Brandon et al., 2024; Wu & Tu, 2024; Liao & Vargas, 2024; Wu & Tu, 2024), which hinders seamless plug-and-play integration. Our focus lies in methods that do not require retraining, with MiniCache (Liu et al., 2024a) serving as a representative approach. By taking advantage of the similarity between the KV pairs across layers, MiniCache combines the cache of every two layers through spherical interpolation, effectively compressing KV cache across layers(see Figure 1(b)). However, MiniCache operates under the implicit assumption that all layers within the merged set contribute equally, which may not always hold true. In fact, research on layer sparsity (Gromov et al., 2024) shows that importance levels vary across layers within the same model, indicating that their contributions may differ.

To investigate this character for the attention layer, we conducte preliminary experiments (Section 4) and identified three key findings: (1) *Certain layers in long-context LLMs exhibit "lazy" behavior*, primarily focusing on semantically unimportant tokens (e.g., the initial few tokens) and the most recent ones during answer generation. (2) *Lazy layers are less important than non-lazy layers w.r.t. long-context capability*: trimming KV cache in non-lazy layers significantly degrades model performance, whereas trimming KV cache in lazy layers has relatively little impact; and (3) After analyzing attention weight patterns, we find that *layer behavior is consistent across tokens for a given input*, and *lazy layers can be easily identified*.

The appearance of lazy layers suggests that we can directly reduce the KV cache for these layers without altering the cache of non-lazy layers or merging cache across layers. Building on this insight, we propose *SimLayerKV*, a simple yet effective method for inter-layer KV cache reduction. This dynamic, selective reduction in KV cache decreases the number of layers requiring cache retention, thereby enhancing computational efficiency. Specifically, we analyze the attention allocation patterns in each layer to determine whether it qualifies as a lazy layer. We then trim the KV cache in lazy layers while retaining the full KV cache in non-lazy layers (see Figure 1(c)). We conduct extensive experiments on three representative LLMs (i.e., LLama2-7B-chat (Touvron et al., 2023), LLama3-8B-Instruct (Dubey et al., 2024), and Mistral-7B-Instruct (Jiang et al., 2023)) across 16 tasks from LongBench (Bai et al., 2023). The results demonstrate that SimLayerKV achieves a KV cache compression ratio of 5$\times$ with only a 1.2% drop in performance when combined with a 4-bit quantization (Liu et al., 2024c). Meanwhile, it integrates seamlessly into popular inference frameworks with just seven lines of code. Additionally, we evaluate SimLayerKV on the Ruler (Hsieh et al., 2024) datasets using Mistral-7B-Instruct, focusing on tasks like Needle-in-a-Haystack (NIAH) and scaling the context length from 4K to 32K, where it performed strongly. Even with input texts at 32K, performance only dropped by 4.4%. The contributions of this work are summarized as follows:

- We observe the phenomenon of lazy layers in long-context LLMs and propose two strategies for identifying them at either the prefilling or decoding stage.

- We introduce SimLayerKV, a simple yet effective method for reducing inter-layer KV cache redundancies that can be implemented with only seven lines of code.
- Our SimLayerKV achieves a KV cache compression ratio of 5× with only a 1.2% drop in performance on the LongBench benchmark on three representative LLMs.

## 2 Related work

Due to the autoregressive architectures of transformer-based LLMs, the key and value states of previously generated tokens can be stored as the KV cache, which facilitates the generation of subsequent tokens without redundant computations. However, despite its benefits, caching introduces a significant bottleneck during inference as it must reside in GPU memory. Several works (Prabhu et al., 2024; Kwon et al., 2023; Lin et al., 2024; Ye et al., 2024) have focused on optimizing KV cache memory at the system level. Other research has investigated reducing KV cache memory requirements by modifying model architectures (Shazeer, 2019; Brandon et al., 2024; Goldstein et al., 2024; Nawrot et al., 2024; Wang et al., 2024a; Yu et al., 2024). For example, grouped-query attention (GQA) (Ainslie et al., 2023) divides the query heads into multiple groups, with each sharing its own set of keys and values. However, these techniques typically need to be applied during pre-training, which can be resource-intensive.

A different line of research focuses on reducing the KV cache memory usage post pre-training. Some techniques (Xiao et al., 2023; Li et al., 2024; Wang et al., 2024a; Zhang et al., 2024b; Liu et al., 2024b; Yang et al., 2024b; Zhang et al., 2024a) identify redundant tokens within each attention layer and evict their associated KV cache, thereby effectively lowering memory usage. Other methods (Hooper et al., 2024; Dong et al., 2024a; Yang et al., 2024c; Dong et al., 2024b; Kang et al., 2024; Sheng et al., 2023) reduce memory consumption by quantizing KV cache from full precision to lower bit values. However, these methods primarily exploit intra-layer KV cache redundancies while overlooking those across layers. These techniques are orthogonal to our approach and can potentially be combined for further improvements.

A distinct line of research (Rajput et al., 2024; Brandon et al., 2024; Wu & Tu, 2024; Liao & Vargas, 2024; Wu & Tu, 2024; Liu et al., 2024a), more closely aligned with our focus, explores the inter-layer KV cache redundancies. For instance, CLA (Brandon et al., 2024) reduces overall KV cache storage by reusing the KV cache from the current layer in subsequent layers. Mix Attention (Rajput et al., 2024) integrates cross-layer cache sharing with sliding window attention, which retains only a small subset of recent tokens in the KV cache, thereby further reducing memory usage. However, these methods require additional training, which is computationally demanding. In contrast, MiniCache (Reid et al., 2024) offers a tuning-free solution by merging every two adjacent layers through spherical interpolation, assuming equal contribution from all layers within the merged set. Our SimLayerKV approach differs by selectively trimming lazy layers, based on the observation that not all layers contribute equally to the overall generation.

## 3 Preliminary

Before introducing SimLayerKV, we formalize our notation and provide a brief overview of the generative inference in autoregressive LLMs, which is the key background knowledge for our method. We denote the input prompt $X = \{x_0, \cdots, x_{m-1}\}$, representing a sequence of tokens, where $m$ is the number of tokens in the input prompt, indicating the sequence length. The total number of tokens, including both the input prompt and the generated responses, is denoted as $n$. The key and value cache for token $x_i$ are represented by $K_{x_i}$ and $V_{x_i}$, respectively.

**Inference stages.** The typical generative LLM inference process involves two stages: (1) *Prefilling*: the autoregressive LLM processes the input prompt $X$ by parallel computing, and also saves the KV cache of each token $x_i \in X$, where $i = 0, 1, \cdots, m-1$. The output of the last token in this stage is the first token $x_m$ of the response. (2) *Decoding*: after the prefilling stage is completed, the LLM generates output tokens $x_j$ one by one, where $j = m+1, m+2, \cdots$, and saves their KV cache. In each decoding step, a new token $x_j$ is generated based on the current token $x_{j-1}$ and the KV Cache stored from earlier steps, continuing until a stop criterion is met.

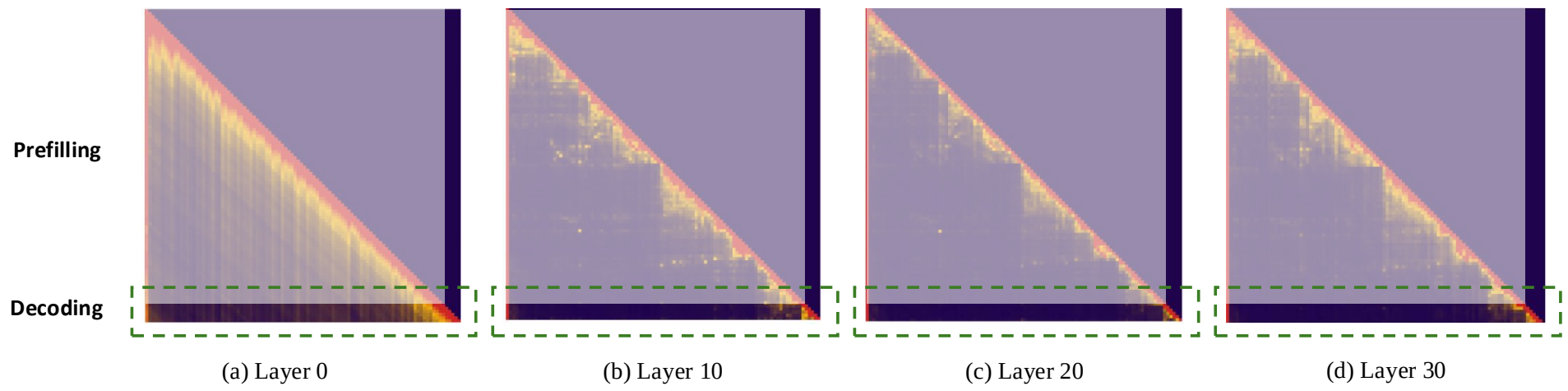

Figure 2: Attention patterns during long-context generation in layers 0, 10, 20, and 30 of the LLaMA3-8B-Instruct model. The green dashed box outlines the decoding stage. Notably, in certain layers (e.g., 20), the model predominantly focuses its attention on initial tokens and recent tokens during the decoding stage, a behavior we identify as characteristic of lazy layers.

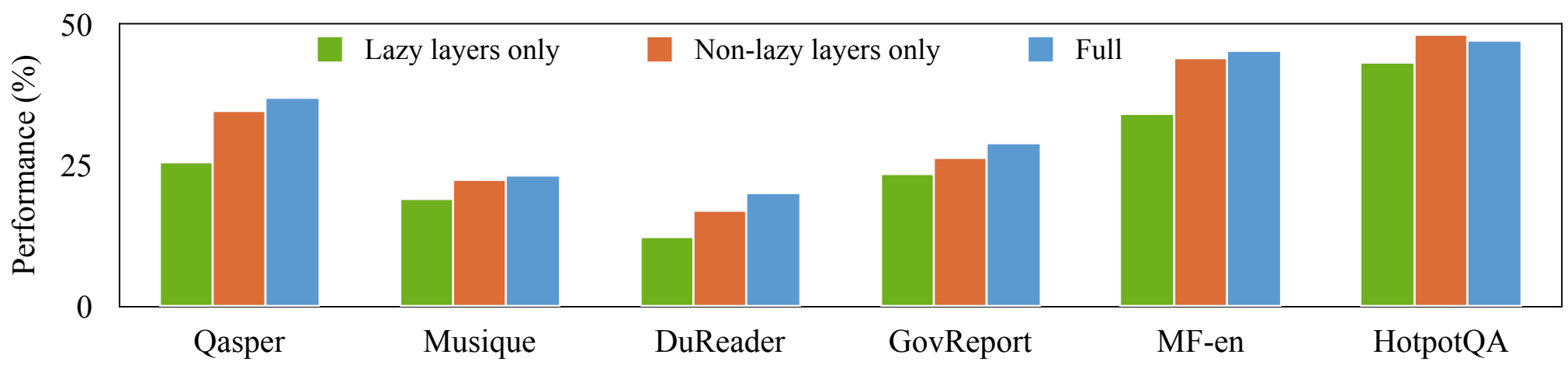

Figure 3: Comparison of the importance of KV cache in lazy and non-lazy layers using LLama3-8B-Instruct. Performance is evaluated across three settings: 1) lazy layers only: trimming KV cache in non-lazy layers, 2) non-lazy layers only: trimming KV cache in lazy layers, and 3) full: using the full KV cache for generation.

## 4 OBSERVATIONS

In this section, we analyze the attention patterns during the prefilling and decoding phase in long-context LLMs, providing insights that motivate our approach to reducing KV cache based on the layer-specific roles in attention allocation. The study is conducted on the LLaMA3-8B-Instruct model (Dubey et al., 2024) using random samples from the LongBench (Bai et al., 2023) benchmark. Our key findings are as follows:

**Layer behavior in long context LLMs during decoding.** Previous research (Xiao et al., 2023) has shown that a large portion of attention in LLMs tends to focus on semantically unimportant tokens (e.g., the first few tokens) and the most recent tokens. We refer to this pattern as *lazy* behavior, where the model "takes shortcuts" by primarily attending to the beginning and end of the sequence, similar to someone skimming a paper by only reading the first few words in the abstract and the conclusion. Although this phenomenon is also known as "attention sink" (Xiao et al., 2023), we choose to call it "lazy behavior" in our context to better highlight the model's tendency to overlook the middle portions of the sequence, emphasizing the shortcut nature. However, in our experiments (See Table 1 and Table 3), we find that when KV cache are retained for only these tokens across all layers, the long-context capabilities of LLMs degrade sharply. This raises an important question: does this lazy behavior disappear when processing long texts?

Through our analysis, we observe that even when handling long texts, many layers continue to exhibit this lazy behavior during decoding (e.g., about 55% in LLama3-8B-Instruct in LongBench benchmark). Figure 2 presents the attention patterns across four different layers (0, 10, 20, and 30). We observe that some layers (e.g., layer 0) do not follow a clear pattern in attention weight distribution, while others (e.g., 20) show a clear lazy behavior pattern. Based on this observation, we define a *lazy layer* as one that primarily attends to a limited subset of tokens, including both the initial tokens $X_{\text{initial}} = \{x_0, x_1, x_2, x_3\}$ and recent $w$ tokens $X_{\text{recent}}$, while allocating minimal attention to the rest of the tokens in the sequence during decoding stage. Intuitively, this suggests that in these lazy layers, most of the KV cache can be dropped, retaining only the portions the model relies on during its "shortcut" behavior, i.e., $X_{\text{initial}}$ and $X_{\text{recent}}$.

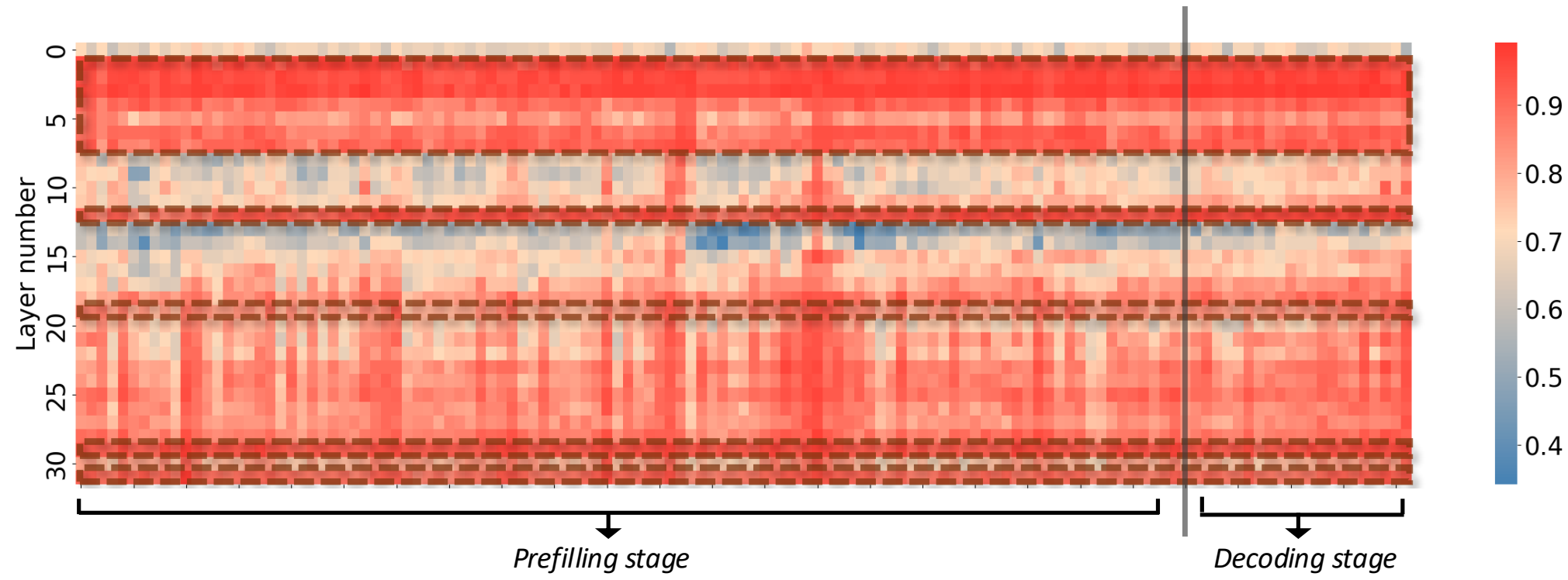

Figure 4: Visualization of attention weights for each token (x-axis) with respect to the **initial** tokens and the **most recent 1024** tokens during the prefilling and decoding stages on LLama3-8B-Instruct, across all layers (y-axis), using a randomly selected sample. Layers with predominantly higher attention on the initial and recent tokens $\{X_{\text{initial}}, X_{\text{recent}}\}$ (indicated by red areas) are referred to as lazy layers. The brown dashed box outlines one such lazy layer.

**Lazy layer is less important than non-lazy layer.** Although attention scores in lazy layers are concentrated on certain tokens, this does not necessarily indicate that these layers are unimportant for long-context capability. To investigate this further, we conduct experiments on 6 random selected tasks from the LongBench benchmark (Bai et al., 2023), including Qasper (Dasigi et al., 2021), Dureader (He et al., 2017), Musique (Trivedi et al., 2022), GovReport (Huang et al., 2021), MultiFieldQA-en (Bai et al., 2023), and HotpotQA (Yang et al., 2018). We test the effect of trimming most of the KV cache, retaining only the cache for $\{X_{\text{initial}}, X_{\text{recent}}\}$ in two scenarios: (1) lazy layers, and (2) non-lazy layers. For a fair comparison, the number of trimmed layers is kept similar in both settings. We also evaluate the vanilla setting, which uses a complete KV cache, for reference.

As shown in Figure 3, trimming the KV cache in non-lazy layers lead to a significant performance drop, with an average decrease of 7.4%. Interestingly, trimming the KV cache in lazy layers results in only an average 1.5% decrease. These results suggest that lazy layers contribute less to the model's overall performance compared to non-lazy layers.

**Layer behavior remains consistent for a given input.** To further explore whether a layer consistently functions as a lazy layer during generation, we visualize the attention weights for $\{X_{\text{initial}}, X_{\text{recent}}\}$ across all layers for all generated tokens in Figure 4, using a randomly selected sample (additional examples are provided in Figure 7). Notably, for a given input prompt, layers that exhibit lazy behavior maintain this pattern relatively consistently across tokens. This suggests a certain degree of stability in attention dynamics throughout the generation process.

## 5 Methodology: SimLayerKV

In this section, we introduce our method SimLayerKV for reducing inter-layer KV cache usage in LLMs by leveraging the concept of *lazy layers* to optimize memory efficiency across layers. Empirical observations in Section 4 reveal that in certain layers, LLMs tend to take shortcuts by predominantly allocating attention weights to the initial and most recent tokens, denoted as $X_{\text{initial}}$ and $X_{\text{recent}}$, respectively. We refer to these layers as lazy layers because they contribute less to modeling long-range dependencies compared to non-lazy layers. Notably, whether a layer functions as lazy remains relatively consistent given a specific input sequence. This consistency suggests that attention patterns can be predicted from the allocation during the generation of previous tokens, enabling early identification of lazy layers in the generation process.

Based on our observations of lazy layers, we aim to optimize memory usage by trimming the KV cache in these layers. Some existing approaches have attempted to optimize attention mechanisms at different layers. For instance, Gemma 2 (Team et al., 2024) employs a predefined mixture of full attention and sliding window attention across different layers during training, treating certain layers as lazy layers. However, this approach relies on a fixed, predefined structure and lacks adaptability

to the input data. In contrast, our method dynamically identifies lazy layers based on their attention allocation patterns, without the need for additional tuning or predefined settings. This dynamic identification allows our model to more flexibly optimize KV cache usage, adapting to different input data more efficiently. Our approach consists of two components: identifying the function of each layer (i.e., whether a layer is lazy) and trimming the KV cache in those identified lazy layers.

### 5.1 IDENTIFYING THE LAYER FUNCTION

To apply SimLayerKV, the first step is to identify which layers function as lazy layers based on their attention allocation patterns. Once these layers are identified, we can proceed to trim their KV cache to optimize memory usage. In the following, we detail our strategies for identifying the layer function. Corresponding to the two stages of the inference process (i.e., prefilling and decoding), we propose two different identification strategies.

1) *Last tokens in prefilling*: We analyze the attention weight allocation when processing the last $w_{\text{last}}$ processed tokens $X_{\text{last}} = \{x_{m-w_{\text{last}}+1}, \cdots, x_m\}$ to identify lazy layers during prefilling. For each layer $l$, we calculate the average attention weights directed toward the $X_{\text{initial}}$ and $X_{\text{recent}}$ for all tokens in $X_{\text{last}}$. If this average exceeds a predefined threshold $\delta$, we classify the layer $l$ as lazy; otherwise, it is considered non-lazy. This can be formalized as:

$$\text{Function}[l] = \begin{cases} \text{lazy layer}, & \text{if } \frac{1}{w_{\text{last}}} \left( \sum_{\hat{x} \in X_{\text{last}}} \left( \sum_{x \in \{X_{\text{initial}}, X_{\text{recent}}\}} A_l(\hat{x}, x) \right) \right) > \delta, \\ \text{non-lazy layer}, & \text{otherwise}, \end{cases} \tag{1}$$

where $A_l(\hat{x}, x)$ represents the attention weight from token $\hat{x}$ to token $x$ in layer $l$ and the threshold $\delta$ is a predefined hyper-parameter.

2) *First token in decoding*: We assess the attention weight distribution when generating the first token $x_{m+1}$ during the decoding phase to identify lazy layers. Specifically, for each layer $l$, if the attention weights directed toward $\{X_{\text{initial}}, X_{\text{recent}}\}$ when generating $x_{m+1}$ exceed $\delta$, we classify the layer as lazy; otherwise, it is not considered lazy. This can be formalized as:

$$\text{Function}[l] = \begin{cases} \text{lazy layer}, & \text{if } \sum_{x \in \{X_{\text{initial}}, X_{\text{recent}}\}} A_l(x_{m+1}, x) > \delta, \\ \text{non-lazy layer}, & \text{otherwise}. \end{cases} \tag{2}$$

**Remark.** During the prefilling stage, flash attention (Dao, 2023) is commonly used to accelerate computations. However, flash attention does not return explicit attention weights, making it challenging to apply the lazy layer identification strategy without recomputing the attention scores, which would introduce additional computational overhead. In contrast, during the decoding stage, tokens are generated one at a time without using flash attention, so the attention weights are readily available. This allows us to apply our identification strategy without extra computation. In our experiment (See Table 6), we find the performance of the two strategies is comparable, with no significant differences.

### 5.2 CACHE STRATEGY

Once lazy layers have been identified, we proceed to trim the KV cache for these specific layers. Lazy layers are characterized by their significant attention allocation to a limited subset of tokens, namely $\{X_{\text{initial}}, X_{\text{recent}}\}$. Thus we retain only the KV cache corresponding to these tokens within lazy layers. This selective retention strategy is similar to approaches used in methods like Gemma 2 (Team et al., 2024), which also retain KV cache for recent tokens in predefined layers.

Specifically, for any lazy layer $l$, we trim its KV cache by retaining only those of tokens in $\{X_{\text{initial}}, X_{\text{recent}}\}$. Otherwise, we retain the full cache. This process can be expressed as:

$$\text{Cache}[l] = \begin{cases} \{K_{\text{initial}}, V_{\text{initial}}, K_{\text{recent}}, V_{\text{recent}}\}, & \text{if Function}[l] = \text{lazy layer}, \\ \text{full KV}, & \text{otherwise}, \end{cases} \tag{3}$$

where Cache$[l]$ represents the KV cache for layer $l$.

Table 1: Performance comparison of SimLayerKV and baseline methods on LLaMA-2-7B-chat, LLaMA-3-8B-Instruct, and Mistral-7B-Intruct using LongBench. **Bold** denotes the best method, and the second best if the top method is Full KV.

| | Single-Doc. QA | | | Muti.-Doc. QA | | | Summary | | | Few-shot | | | Syn. | | Code | | |
|---|---|---|---|---|---|---|---|---|---|---|---|---|---|---|---|---|---|
| | NrtvQA | Qasper | MF-en | HotpotQA | Musique | DuReader | GovReport | QMSum | MultiNews | TREC | TriviaQA | SAMSum | PCount | PRe | LCC | RB-P | **Average** |
| ***LLaMA2-7B-chat*** | | | | | | | | | | | | | | | | | |
| Full | **18.5** | **18.3** | **36.4** | 26.3 | 7.6 | 7.9 | **26.9** | **21.0** | 26.0 | **64.0** | 83.2 | **41.1** | **4.5** | **7.0** | 59.9 | 54.7 | **31.5** |
| Str. | 13.0 | 12.6 | 26.7 | 23.5 | 4.5 | 4.4 | 21.1 | 19.9 | 24.2 | 61.0 | 82.8 | 38.9 | 3.5 | **3.5** | 59.0 | 52.2 | 28.2 |
| Mini. | 13.1 | 13.3 | 27.5 | 14.9 | 4.1 | **9.8** | 21.5 | **20.9** | 24.3 | 63.0 | 83.1 | 35.1 | 3.8 | **3.5** | 53.4 | 46.5 | 27.4 |
| +Q. | 16.4 | 13.9 | 29.4 | 14.1 | 3.9 | 9.7 | 21.4 | 20.5 | 24.4 | 61.5 | 79.1 | 31.1 | 2.3 | 1.0 | 53.1 | 46.2 | 26.7 |
| Ours | **18.4** | **17.3** | 30.9 | 27.3 | 7.7 | 7.2 | 26.3 | 20.4 | **26.3** | **64.0** | **83.5** | **40.7** | 2.5 | 2.0 | **60.3** | **54.9** | **30.6** |
| +Q. | 17.3 | 16.5 | **31.5** | **27.7** | **8.5** | 6.9 | **26.6** | 20.5 | **26.3** | 62.5 | 81.8 | 39.8 | **4.0** | 2.5 | 57.5 | 51.9 | 30.1 |
| ***LlaMA-3-8B-Instruct*** | | | | | | | | | | | | | | | | | |
| Full | 23.4 | **36.9** | **45.2** | 47.0 | **23.1** | **20.1** | **28.8** | **23.3** | **27.0** | 73.5 | **90.6** | **42.0** | 3.5 | 72.0 | 58.1 | 51.3 | **41.6** |
| Str. | 19.5 | 23.8 | 28.5 | 40.5 | 16.8 | 12.1 | 22.8 | 21.4 | 25.4 | 66.0 | 86.6 | 40.2 | 3.5 | 72.0 | 59.7 | **54.2** | 37.1 |
| Mini. | 18.8 | 30.3 | 31.6 | 36.2 | 18.6 | 15.9 | 23.8 | 20.1 | 25.5 | **74.5** | 84.5 | 37.4 | **4.9** | 64.8 | 48.5 | 45.3 | 36.3 |
| +Q. | 17.5 | 28.3 | 30.8 | 35.9 | 19.0 | 15.9 | 23.9 | 19.6 | 25.8 | 73.5 | 84.2 | 36.8 | 4.5 | 65.3 | 49.1 | 45.3 | 35.9 |
| Ours | **23.6** | **34.7** | **43.9** | **48.0** | **22.5** | 17.0 | **26.2** | 22.5 | **26.2** | 73.5 | 89.3 | **40.6** | 3.5 | **72.5** | 58.0 | 50.7 | **40.8** |
| +Q. | **23.6** | 33.6 | 42.5 | 45.4 | 21.8 | **17.3** | 25.8 | **23.0** | 26.0 | 72.4 | **89.6** | 40.3 | 3.2 | 70.6 | **60.0** | 49.8 | 40.3 |
| ***Mistral-7B-Instruct*** | | | | | | | | | | | | | | | | | |
| Full | **29.3** | **41.1** | 54.8 | 43.8 | 26.8 | 32.3 | **33.8** | **24.3** | **28.0** | **74.0** | **88.4** | **47.2** | **3.5** | 63.0 | 61.4 | **61.8** | **44.6** |
| Str. | 21.3 | 27.5 | 31.7 | 39.5 | 17.9 | 17.7 | 24.3 | 20.5 | 25.6 | 67.5 | 87.0 | 45.5 | **3.5** | 54.0 | 61.8 | 58.9 | 37.8 |
| Mini. | 22.2 | 32.1 | 44.8 | 41.7 | 23.0 | 20.3 | 24.8 | 21.3 | 26.0 | 65.0 | 86.7 | 40.4 | **3.5** | 46.0 | 52.8 | 47.9 | 37.4 |
| +Q. | 22.2 | 31.4 | 42.8 | 41.0 | 22.8 | 20.1 | 24.4 | 21.6 | 25.9 | 66.0 | 86.3 | 40.2 | **3.5** | 47.0 | 52.4 | 47.4 | 37.2 |
| Ours | 25.0 | 37.7 | 56.4 | 43.7 | 26.4 | **33.5** | **33.1** | 23.4 | **27.4** | **74.0** | 88.1 | **47.1** | **3.5** | **64.5** | **62.3** | **61.3** | **44.2** |
| +Q. | **25.1** | **38.7** | **56.5** | **44.4** | **27.2** | 31.0 | 31.6 | **23.7** | 27.1 | 73.9 | **88.4** | 46.4 | **3.5** | 61.0 | 60.3 | 60.0 | 43.7 |

## 6 Experiments

In this section, we empirically validate that SimLayerKV can accelerate decoding while maintaining long-text capabilities and uncover several insightful findings.

### 6.1 Settings

**Baselines.** To evaluate the effectiveness of our proposed SimLayerKV, we compare it against the following baselines: 1) Full KV (Full): A method that retains KV cache for all tokens at each layer during generation. 2) Streaming LLM (Str.) (Xiao et al., 2023): An intra-layer KV cache reduction technique that keeps only the KV cache for the first four tokens and the most recent $w$ tokens at each attention layer during generation. 3) MiniCache (Mini.) (Liu et al., 2024a): An inter-layer KV cache reduction method that merges KV cache of every two adjacent layers after the model's midpoint using spherical interpolation while retaining important tokens to reduce cache storage. Additionally, for both MiniCache and our SimLayerKV, we evaluate their performance when combined with 4-bit quantization (Liu et al., 2024c) to assess their compatibility with quantization techniques.

**Datastes and evaluation metrics.** To evaluate SimLayerKV's performance on tasks with long-context inputs, we test it on the LongBench benchmark (Bai et al., 2023) and compare the results with baseline methods. LongBench is a multi-task benchmark designed to assess the long-context capabilities of LLMs, consisting of datasets that span various tasks such as single-document QA (Kočiskỳ et al., 2018; Dasigi et al., 2021), multi-document QA (Yang et al., 2018; Ho et al., 2020; Trivedi et al., 2022; He et al., 2017), summarization (Huang et al., 2021; Zhong et al., 2021; Fabbri et al., 2019; Wu et al., 2023), few-shot learning (Joshi et al., 2017; Gliwa et al., 2019; Joshi et al., 2017; NLPCC, 2014), synthetic tasks (Raffel et al., 2020), and code generation (Guo et al., 2023; Liu et al., 2023). For evaluation, we use the metrics recommended by LongBench. Additionally, we provide the compression ratios for both the number of layers and memory usage of the KV cache. For layers, the ratio is calculated as the total number of layers divided by the number of

layers with reduced KV cache. For the KV cache, the ratio is the original memory usage divided by the memory usage after compression. Due to space constraints, we only include the performance of 16 randomly selected tasks out of the 21 LongBench tasks in the main text. The performance on the remaining 5 tasks is provided in Appendix A.3 Table 9.

We also evaluate whether SimLayerKV can preserve in-context retrieval capabilities while trimming KV cache in lazy layers. The evaluation is conducted on the Needle-In-A-Haystack (NIAH) benchmark (Kamradt, 2023) including various types and quantities of needles, along with tasks such as aggregation for common/frequent words, question answering (QA), and multi-hop variable tracing (VT), all provided by the Ruler benchmark (Hsieh et al., 2024). We report the performance of Mistral-7B-Instruct with input context lengths of 4K, 8K, 16K, and 32K. The evaluation is conducted using the metrics recommended by Ruler.

**Implementation details.** Our experiments are based on widely used LLMs, specifically LLaMa2-7B-chat (Touvron et al., 2023), LLaMa3-8B-Instruct (Dubey et al., 2024), and Mistral-7B-Instruct (Jiang et al., 2023). The input context window sizes are 4K, 8K, and 32K, with average tokenized sequence lengths of approximately 13K, 10K, and 12K in LongBench. It is worth noting that we do not use different thresholds for each task. Instead, we search for the optimal threshold based on the synthetic Need-in-a-Haystack task and apply the same threshold across all tasks in different benchmarks. The thresholds ($\delta$) for the models are 0.65, 0.9, and 0.8 respectively. We adopt a generative format where answers are produced using greedy decoding for all tasks. We chose the first token identification strategy during the decoding stage in our experiments. For MiniCache, as the code was not open-sourced before our submission, we reimplemented it based on the original paper and the SLERP (Shoemake, 1985) code it references. We followed all the hyper-parameters outlined in the paper, except for the number of retention tokens. To ensure a fair comparison, we set the number of retention tokens to 1024, matching the window size $w$ used in our SimLayerKV method. Note that even with the same retention window size, MiniCache's compression ratio is still lower than that of our SimLayerKV as shown in Table 2. All the experiments are conducted using NVIDIA A100.

### 6.2 Experiments on LongBench

Table 1 summarizes the performance across various tasks in the LongBench (Bai et al., 2023) benchmark, and Table 2 shows the corresponding compression ratio. We have the following findings:

Table 2: Comparison Ratio of layer and KV cache memory on LongBench. The higher the ratio, the better the performance in terms of compression efficiency. **Bold** denotes the method with the highest compression ratio.

| | LLaMA2-7B | | LLaMA-3-8B | | Mistral-7B | |
|---|---|---|---|---|---|---|
| | Layers | KV | Layers | KV | Layers | KV |
| MiniCache | 1.33× | 1.27× | 1.33× | 1.25× | 1.33× | 1.26× |
| + 4bit Q. | 1.33× | 3.95× | 1.33× | 3.88× | 1.33× | 3.92× |
| SLKV(ours) | **1.39**× | 1.35× | **2.04**× | 1.85× | **1.83**× | 1.71× |
| + 4bit Q. | 1.35× | **4.11**× | 1.96× | **5.57**× | 1.81× | **5.26**× |

**LLMs exhibit redundancy across layers.** Table 2 demonstrates that MiniCache and our SimLayerKV achieve average layer compression ratios of 1.33× and 1.75×, respectively. Our SimLayerKV demonstrates notably higher compression ratios in models with strong long-context capabilities (i.e., LLaMA-3-8B-Instruct and Mistral-7B-Instruct) than in those with weaker ones (i.e., LLaMA-2-7B-chat). Meanwhile, as indicated in Table 1, while MiniCache shows some limitations, our SimLayerKV allows the model to continue effectively managing long-text tasks with minimal loss in performance (i.e., an average drop of 0.7%). After integrating 4-bit quantization, our SimLayerKV achieves a remarkable compression rate of 4.98× on average, while still maintaining robust performance. Compared to SimLayerKV without quantization, the average performance drop is only 0.5%.

**SimLayerKV outperforms MiniCache on average.** Unlike MiniCache, our approach does not rely on complex interpolation and retention strategies to merge KV cache from different layers. Instead, we simply identify lazy layers based on the attention weight patterns and trim the KV cache in those layers. Additionally, our method seamlessly integrates reduction into the decoding process. More importantly, as shown in Table 1 and Table 2, our results show a clear advantage over MiniCache, whether or not combined with quantization, achieving 4.8% higher performance and a 1.29× greater KV cache compression ratio, further emphasizing the efficiency and effectiveness of our approach.

Table 3: Performance comparison of SimLayerKV and baseline methods on Ruler benchmark using Mistral-7B-Instruct. NIAH: Needle-In-A-Haystack, S: Single Key, MK: Multi-Keys, MV: Multi-Values, MQ: Multi-Queries, CWE: Common Words Extraction, FWE: Frequent Words Extraction, QA: Question Answering, VT: Variable Tracking. **Bold** denotes the best method, and the second best if the top method is Full KV.

| Context Length | Method | Retrieval: NIAH | | | | Aggregation | | QA | VT | Avg. |
|---|---|---|---|---|---|---|---|---|---|---|
| | | S | MK | MV | MQ | CWE | FWE | | | |
| 4096 | Full | **99.9** | **99.4** | 87.2 | **99.3** | **99.5** | 85.9 | **64.1** | **99.4** | **91.8** |
| | MiniCache | 37.2 | 18.1 | 20.6 | 30.9 | 77.3 | 77.4 | 55.8 | 77.8 | 49.4 |
| | SimLayerKV | **99.7** | **99.4** | **87.6** | **84.0** | **98.9** | **86.9** | **63.6** | **98.5** | **89.8** |
| 8192 | Full | **99.9** | 98.5 | **79.5** | **97.9** | **95.4** | **76.1** | **61.8** | **98.3** | **88.4** |
| | MiniCache | 21.6 | 5.3 | 7.9 | 12.4 | 31.0 | 53.8 | 46.0 | 55.0 | 29.1 |
| | SimLayerKV | **99.8** | **98.6** | **79.0** | **89.1** | **87.8** | **76.1** | **60.4** | **95.0** | **85.7** |
| 16384 | Full | **99.9** | **95.1** | **81.8** | **96.3** | **89.4** | **96.9** | **58.8** | **94.1** | **89.0** |
| | MiniCache | 14.0 | 1.2 | 3.1 | 3.1 | 15.9 | 49.3 | 38.3 | 34.0 | 19.9 |
| | SimLayerKV | **99.8** | **94.8** | **81.8** | **90.5** | **73.4** | **89.3** | **57.4** | **90.5** | **84.7** |
| 32768 | Full | 96.6 | **78.9** | **87.0** | **93.9** | **75.1** | **93.3** | 51.2 | **92.4** | **83.5** |
| | MiniCache | 5.5 | 0.7 | 0.5 | 0.8 | 7.5 | 20.3 | 30.5 | 22.1 | 11.0 |
| | SimLayerKV | **96.7** | **78.2** | **86.2** | **91.1** | **48.6** | **88.5** | **52.1** | **91.7** | **79.1** |

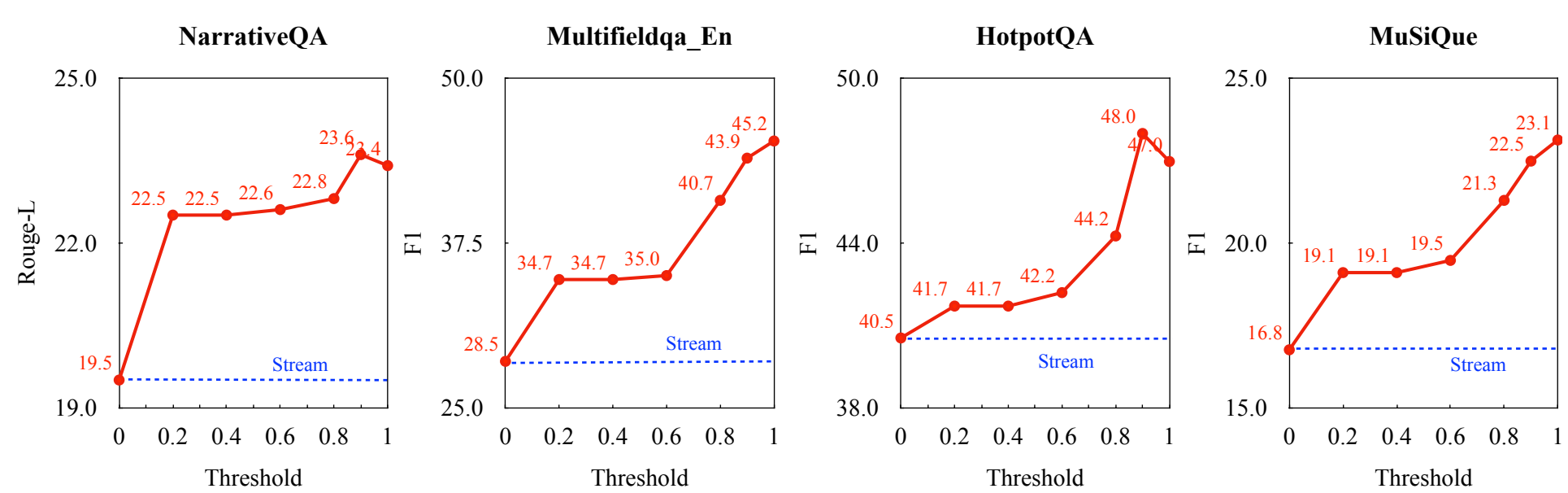

Figure 5: Effect of threshold $\delta$ on lazy layer identification using LLama3-8B-Instruct: Increasing the threshold results in more layers being identified as non-lazy rather than lazy.

### 6.3 Experiments on Ruler

Table 3 summarizes the performance across various tasks in the Ruler (Hsieh et al., 2024) benchmark, with the context length ranging from 4K to 32K. We find that SimLayerKV maintains strong performance on the Single Key, Multiple Keys, and Multiple Values Needle-In-A-Haystack (NIAH) tasks, exhibiting minimal to no degradation. For example, even with a 32K input context, SimLayerKV results in only a slight performance drop of 0.47% compared to the full KV cache. Our method also performs well on the Question Answering and Variable Tracking tasks, which involve long context capabilities similar to NIAH. However, we observe a performance drop (8.2% on average) on the Mutliple Queries NIAH with SimLayerKV. This may be due to the data-dependent nature of lazy layer identification in our approach. Ideally, varying the number of queries should lead to different layers being identified as lazy and reduced accordingly, but currently, the same layers are reduced regardless of the query count. We also observe a similar phenomenon in aggregation tasks. Although the Common Words Extraction (CWE) and Frequent Words Extraction (FWE) tasks are quite similar, both aiming to return the top-$K$ frequent words in the context, our method shows a significantly more pronounced decline in performance on CWE. One possible reason is that, in the FWE task, the value of $K$ is consistently fixed at 3, while in the CWE task, $K$ increases with the context length, making the task progressively more challenging for our method. In addition, we measure throughput under the maximum batch size for input sequence lengths of 4K, 8K, 16K, and 32K using LLaMA-3-8B. The throughput (tokens/s) for SimLayerKV relative to the Full method was 1.44×, 1.78×, 2.17×, and 1.75×, respectively, suggesting our method can increase the throughput effectively.

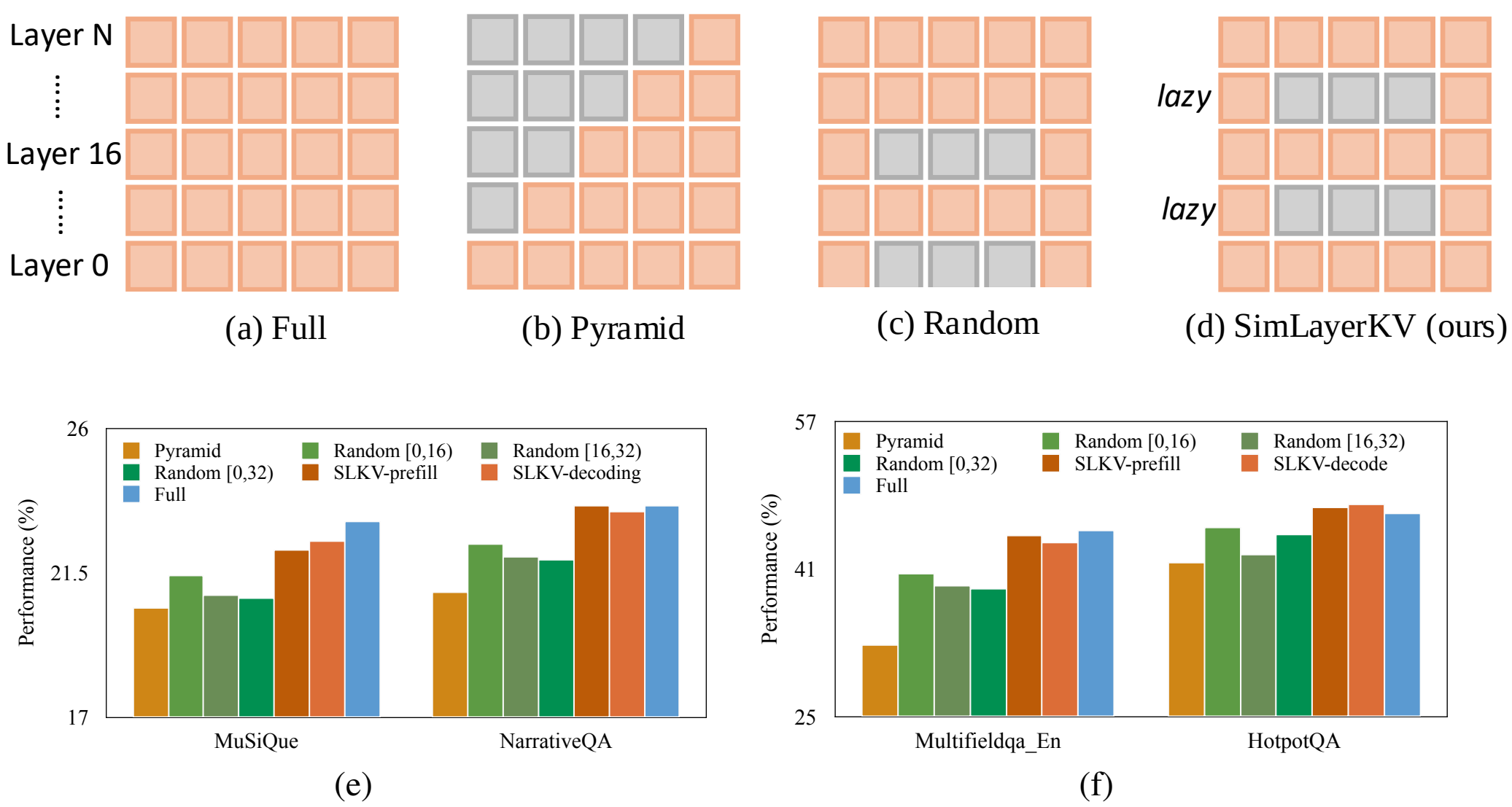

Figure 6: Different strategies for dropping KV cache at the layer level and their performance on LLama3-8B-Instruct: 1) Full: Use full KV cache for all layers. 2) Pyramid: KV cache are progressively reduced as the layers increase, forming a pyramid-like structure. 3) Random: Drop the KV cache in randomly selected layers within the ranges $[0, 16)$, $[16, 32)$, and $[0, 32)$. 4) Our SimLayerKV (SLKV): Identify lazy layers during either the prefilling or decoding stages, and trim the KV cache accordingly. We keep a **same** number of dropped KV cache for all strategies, except Full.

### 6.4 Ablation studies & analysis

**Impact of threshold on lazy layer identification.** To assess the impact of the threshold $\delta$ in identifying lazy layers, we conduct an ablation analysis using the LLama3-8B-Instruct model, varying $\delta$ from 0, 0.2, up to 1. As illustrated in Figure 5, we observe that as the threshold increases, the model's performance shows little to no change or only slow improvement initially. However, after exceeding 0.6, the performance improves rapidly, and by 0.9, it approaches the performance seen when the threshold equals 1 in most tasks. This indicates that as the threshold increases, the likelihood of accurately identifying and trimming truly lazy layers increases, allowing the model to maintain high performance while reducing unnecessary computations.

**Effect of different strategies for dropping KV cache at layer level.** As shown in Figure 6 (a-d), we experiment with four different strategies. We ensured the same number of dropped KV cache for each strategy, except for Full. The results shown in Figure 6 (e-f) indicate significant reductions for Pyramid and Random strategies, suggesting that the predefined expectations about each layer's function may not fully align with their actual roles. Moreover, the performance difference between SLKV-prefill and SLKV-decode strategies is minimal, with only slight reductions compared to the full KV cache (0.20% and 0.28% on average, respectively). This indicates that both approaches are effective in reducing cache usage while maintaining performance, regardless of whether lazy layers are identified during the prefilling or decoding stages.

## 7 Conclusion

In this work, we introduced SimLayerKV, a simple yet effective method for compressing the KV cache in LLMs. By identifying lazy layers and trimming their KV cache, SimLayerKV effectively reduced inter-layer KV cache redundancies. Experiments on three different LLMs across 16 datasets from the LongBench benchmark demonstrated that SimLayerKV, with only seven lines of code, achieves a KV cache compression ratio of $5\times$ with only a 1.2% drop in performance when combined with 4-bit quantization. For future work, we aim to combine our inter-layer KV cache compression method, SimLayerKV, with other powerful intra-layer compression methods like H2O (Zhang et al., 2024b) to further enhance performance and efficiency.

## Reproducibility statement

We have taken several steps to ensure the reproducibility of our results. Detailed descriptions of the experimental setup, including hyper-parameters, base models, and datasets, are provided in Section 6.1. Meanwhile, both the datasets and base models used in our experiments are open-sourced and readily available. Additionally, we provide an anonymous source code in the supplemental materials.

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

# A APPENDIX

## A.1 LIMITATION

While our SimLayerKV has demonstrated significant advantages in inter-layer KV cache compression, we have primarily focused on combining it with quantization, as quantization is one of the most widely used techniques. However, there are many other KV cache optimization methods, such as intra-layer eviction, which are orthogonal to our approach. In this study, we have not explored the potential of integrating our method with these techniques. In the future, we aim to combine our method with other optimization strategies, to further improve performance and efficiency. This will help validate the effectiveness of our method in a broader framework and potentially lead to even greater performance gains. Meanwhile, for simplicity, we have only explored KV cache redundancies across layers in this work. In the future, we plan to extend our approach to consider redundancies across attention heads as well.

## A.2 PSEUDO CODE

The pseudo-code for SimLayerKV-prefill and SimLayerKV-decoding are in Table 4 and Table 6 respectively. In addition, we also provide the pseudo-code for SimLayerKV-prefill with flash attention in Table 5. In our experiments, the reduction in throughput compared to the original (assumed to be 1) is neglectable — between 0.0058 and 0.0014, depending on the sequence length (with longer sequences experiencing smaller reductions, in the range of 4K to 32K tokens).

Table 4: Pseudo code in torch style for our SimLayerKV-prefilling.

```
def SLKV_prefilling(
  query_states, # batch_size * num_heads * seq_len * head_dim
  key_states, # batch_size * num_heads * seq_len * head_dim
  value_states, # batch_size * num_heads * seq_len * head_dim
  window_size,
  threshold,
  ):
  attn_weights = compute_attn(query_states, key_states, attention_mask)
  lazy_weights = compute_lazy_weights(attn_weights)
  if lazy_weights ≥ threshold:
    key_states = torch.cat([key_states[:,:,0:4],
                           key_states[:,:,-window_size:]],dim=-2)
    value_states = torch.cat([value_states[:,:,0:4],
                             value_states[:,:,-window_size:]],dim=-2)
  return key_states, value_states
```

## A.3 ADDITIONAL EXPERIMENTS

**Comparision with intra-layer KV cache compression methods & Additional LLMs** We also compare SimLayerKV with the intra-layer KV cache compression method SnapKV (Li et al., 2024), which compresses the KV cache into a fixed length by selecting clustered important KV positions for each attention head based on attention scores. We use two additional LLMs, i.e., Qwen2.5-3B-Instruct (Yang et al., 2024a; Team, 2024) and Yi-1.5-9B-Chat (Young et al., 2024). Note that our SimLayerKV focuses on intra-layer KV cache redundancies while they study inter-layer redundancies, and our approach is orthogonal to them. For the SnapKV method, due to its head-wise KV eviction mechanism, it necessitates storing KV cache for $n_{\mathrm{q}}$ heads instead of the conventional $n_{\mathrm{kv}}$, where $n_{\mathrm{q}}$ is the number of heads for query and $n_{\mathrm{kv}}$ is the number of heads for key and value. For models using the GQA technique, $n_{\mathrm{q}} = g * n_{\mathrm{kv}}$ and $g$ is the group number. For example, in Qwen2.5-3B-Instruct and Yi-1.5-9B-Chat, $g$ is equal to 8. To ensure a fair comparison and create relatively similar conditions for each method, we standardize the size of recent windows $w$ for SnapKV and our SimLayerKV to 768 and 1024 respectively. As shown in Table 7, we can see that our Sim-

Table 5: Pseudo code in torch style for our SimLayerKV-prefilling with flash attention. lse: logsumexp. The additional computation introduced by our SimLayerKV is highlighted in blue box.

```
def SLKV_prefilling_with_flash_attn
  query_states, # batch_size * num_heads * seq_len * head_dim
  key_states, # batch_size * num_heads * seq_len * head_dim
  value_states, # batch_size * num_heads * seq_len * head_dim
  window_size,
  threshold,
  w_last,
  w_recent,
  ):
  attn_out, lse = flash_attn(query_states, key_states, value_states,
                             causal=True, return_lse=True)
  q_last = query_states[:, -w_last:].permute(0, 2, 1, 3)
  k_comb = torch.cat([key_states[:, 0:w_sink], key_states[:, -w_recent:]],
                     dim=1).permute(0, 2, 3, 1)
  log_lazy_weight = torch.matmul(q_last, k_comb).logsumexp(dim=-1) - lse
  if log_lazy_weights ≥ log(threshold):
    key_states = torch.cat([key_states[:,:,0:4],
                            key_states[:,:,-window_size:]],dim=-2)
    value_states = torch.cat([value_states[:,:,0:4],
                              value_states[:,:,-window_size:]],dim=-2)
  return key_states, value_states
```

Table 6: Pseudo code in torch style for our SimLayerKV-decoding.

```
def SLKV_decoding(
  query_states, # batch_size * num_heads * 1 * head_dim
  key_states, # batch_size * num_heads * seq_len * head_dim
  value_states, # batch_size * num_heads * seq_len * head_dim
  window_size,
  threshold,
  ):
  attn_weights = compute_attn(query_states, key_states, attention_mask)
  lazy_weights = (attn_weight[:,:,:,0:4]
                  +attn_weight[:,:,:,-window_size:]).sum(dim=-1).mean(dim=1)
  if lazy_weights ≥ threshold:
    key_states = torch.cat([key_states[:,:,0:4],
                            key_states[:,:,-window_size:]],dim=-2)
    value_states = torch.cat([value_states[:,:,0:4],
                              value_states[:,:,-window_size:]],dim=-2)
  return key_states, value_states
```

LayerKV achieves comparable performance with SnapKV with a slightly higher compression ratio.

**Combination with SimLayerKV and intra-layer method SnapKV.** To illustrate the orthogonality between inter-layer and intra-layer KV cache compression methods, we provide additional experiments combining SimLayerKV with SnapKV. In these experiments, SnapKV is applied to compress the KV cache for non-lazy layers, while SimLayerKV operations are retained for lazy layers. To maintain consistency with Table 7, we use Qwen2.5-3B-chat-32K in this analysis. As shown in Table 8, our SimLayerKV can be combined with the intra-layer KV cache compression method to reduce the KV cache further while maintaining performance. This suggests that SimLayerKV is orthogonal to existing methods that focus on reducing intra-layer KV cache redundancies.

Table 7: Performance comparison of SimLayerKV and intra-layer KV cache compression models on Yi-9B-chat-16K and Qwen2.5-3B-chat-32K using LongBench. SKV: snapKV.

| | Yi-9B-chat-16K | | | | Qwen2.5-3B-chat-32K | | | |
|---|---|---|---|---|---|---|---|---|
| | **Full** | **Str.** | **SKV** | **Ours** | **Full** | **Str** | **SKV** | **Ours** |
| ***Single-Document QA*** | | | | | | | | |
| NrtvQA | 26.1 | 21.3 | 23.0 | 26.0 | 22.6 | 21.8 | 21.6 | 22.1 |
| Qasper | 39.7 | 27.4 | 38.7 | 38.2 | 34.1 | 24.4 | 32.9 | 30.9 |
| MF-en | 43.3 | 28.0 | 41.5 | 42.1 | 44.0 | 27.1 | 42.4 | 43.8 |
| MF-zh | 55.8 | 35.1 | 55.3 | 52.4 | 51.6 | 32.1 | 49.9 | 52.6 |
| ***Multi-Document QA*** | | | | | | | | |
| HotpotQA | 48.2 | 42.3 | 47.9 | 47.0 | 40.4 | 35.4 | 40.5 | 40.1 |
| 2WikiMQA | 39.6 | 35.4 | 40.0 | 39.8 | 38.2 | 36.5 | 38.7 | 37.0 |
| Musique | 26.4 | 21.9 | 25.0 | 25.6 | 16.1 | 12.0 | 16.0 | 16.8 |
| DuReader | 26.4 | 14.9 | 19.6 | 25.4 | 33.7 | 15.5 | 24.1 | 30.2 |
| ***Summarization*** | | | | | | | | |
| GovReport | 33.1 | 14.7 | 27.1 | 32.7 | 31.8 | 22.5 | 22.0 | 28.7 |
| QMSum | 21.7 | 19.6 | 22.2 | 21.6 | 22.9 | 20.6 | 23.0 | 22.8 |
| MultiNews | 25.5 | 19.5 | 23.6 | 25.1 | 24.7 | 22.9 | 22.5 | 23.8 |
| VCSUM | 14.3 | 13.1 | 13.1 | 13.7 | 15.3 | 15.0 | 13.2 | 14.8 |
| ***Few-shot Learning*** | | | | | | | | |
| TREC | 71.0 | 67.0 | 70.4 | 71.5 | 66.5 | 61.0 | 63.0 | 67.0 |
| TriviaQA | 87.7 | 85.7 | 87.3 | 88.0 | 87.2 | 88.0 | 88.1 | 88.2 |
| SAMSum | 42.8 | 40.5 | 40.1 | 41.1 | 44.0 | 42.7 | 43.5 | 44.0 |
| LSHT | 34.5 | 22.0 | 37.0 | 33.3 | 34.0 | 25.5 | 34.0 | 34.0 |
| ***Synthetic Task*** | | | | | | | | |
| PCount | 4.0 | 4.5 | 2.0 | 4.5 | 2.5 | 4.0 | 3.5 | 4.0 |
| PRe | 56.0 | 14.8 | 62.0 | 54.3 | 41.5 | 37.5 | 45.0 | 42.0 |
| PRz | 92.5 | 26.0 | 89.4 | 90.5 | 34.3 | 14.1 | 34.3 | 36.1 |
| ***Code Completion*** | | | | | | | | |
| LCC | 63.4 | 62.9 | 64.5 | 64.0 | 56.9 | 55.4 | 55.1 | 56.8 |
| RB-P | 60.8 | 57.9 | 60.2 | 60.2 | 56.3 | 52.8 | 53.9 | 55.9 |
| **Average** | 43.5 | 32.1 | 42.2 | 42.7 | 37.9 | 35.6 | 36.5 | 37.7 |
| **Compress. Ratio** | 1× | 13.5× | 1.7× | 1.8× | 1× | 9.9× | 1.2× | 1.7× |

**Experiment results on other datasets on LongBench datasets** Due to space constraints, we only included the performance of 16 out of the 21 LongBench tasks in the main text. Experiments result on additional 5 tasks in LongBench datasets can be found in Table 9.

**Comparison with additional baselines.** We added the comparison with SqueezeAttention (Wang et al., 2024b) in the LongBench benchmark using LLaMA3-8B-Instruct. The results in Table 10 indicate that SimLayerKV preserves long-context capabilities better than SqueezeAttention under similar compression ratios. Additionally, SqueezeAttention can not reduce peak memory usage during prefilling.

**Performance on larger models and compression ratio across different datasets.** We conduct additional experiments with LLaMA3-70B-Instruct, and evaluate the compression ratio and corresponding performance of our SimLayerKV (w/o quantization) in tasks from the LongBench benchmark. The results in Table 11 show that lazy layers are more noticeably present in larger models, and our method successfully compresses KV caches while maintaining performance. Furthermore, we observe that the phenomenon of lazy layers is consistent across different datasets.

**Additional ablation studies.** We adopt hyperparameters either directly from StreamingLLM (i.e., $w_{\text{sink}}$ and $w_{\text{recent}}$), ensuring consistency with established practices in the field, or through preliminary

Table 8: Experiment results on combining SimLayerKV and intra-layer KV cache compression method SnapKV using Qwen2.5-3B-chat on LongBench.

| | **Single-Doc. QA** | | | | **Muti.-Doc. QA** | | | | **Summary** | | | |
|---|---|---|---|---|---|---|---|---|---|---|---|---|
| | NrtvQA | Qasper | MF-en | MF-zh | HotpotQA | 2WikiMQA | Musique | DuReader | GovReport | QMSum | MultiNews | VCSUM |
| ***Qwen2.5-3B-chat*** | | | | | | | | | | | | |
| SnapKV | 21.6 | 32.9 | 42.4 | 49.9 | 40.5 | 38.7 | 16.0 | 24.1 | 22.0 | 23.0 | 22.5 | 13.2 |
| SnapKV+SimlayerKV | 20.2 | 32.3 | 43.0 | 50.0 | 48.8 | 37.6 | 20.7 | 22.7 | 21.9 | 22.8 | 22.4 | 12.8 |

| | **Few-shot** | | | | **Syn.** | | | **Code** | | | |
|---|---|---|---|---|---|---|---|---|---|---|---|
| | TREC | TriviaQA | SAMSum | LSHT | PCount | PRe | PRz | LCC | RB-P | **Average** | **Comp. Ratio** |
| ***Qwen2.5-3B-chat*** | | | | | | | | | | | |
| SnapKV | 63.0 | 88.1 | 43.5 | 34.0 | 3.5 | 45.0 | 34.3 | 55.1 | 53.9 | 36.5 | 1.2× |
| SnapKV+SimlayerKV | 65.5 | 87.8 | 43.4 | 39.0 | 4.0 | 43.0 | 41.0 | 57.4 | 53.8 | **37.6** | **1.7×** |

Table 9: Performance comparison of SimLayerKV and baseline methods on LLaMA-2-7B-chat, LLaMA-3-8B-Instruct, and Mistral-7B-Intruct on additional tasks of LongBench.

| | **MF-zh** | **2Wiki.** | **VCSum** | **LSHT** | **PRz** |
|---|---|---|---|---|---|
| ***LLaMA2-7B-chat*** | | | | | |
| Full | 11.3 | 31.4 | 0.2 | 17.3 | 5.0 |
| Str. | 6.7 | 23.1 | 0.2 | 14.8 | 1.0 |
| Mini. | 8.7 | 19.8 | 4.4 | 15.0 | 0.5 |
| +Q. | 8.0 | 18.6 | 3.8 | 13.0 | 0.5 |
| Ours | 9.1 | 31.6 | 0.2 | 17.8 | 4.5 |
| +Q. | 9.3 | 27.6 | 0.2 | 16.0 | 7.0 |
| ***LlaMA-3-8B-Instruct*** | | | | | |
| Full | 56.1 | 35.3 | 14.7 | 23.5 | 94.0 |
| Str. | 35.2 | 29.1 | 12.6 | 20.0 | 23.0 |
| Mini. | 50.3 | 30.1 | 14.7 | 22.5 | 80.4 |
| +Q. | 51.6 | 27.9 | 13.9 | 23.0 | 83.4 |
| Ours | 55.0 | 31.8 | 11.6 | 23.3 | 87.0 |
| +Q. | 56.1 | 33.7 | 13.5 | 24.0 | 89.5 |
| ***Mistral-7B-Instruct*** | | | | | |
| Full | 56.7 | 39.1 | 15.7 | 31.3 | 92.5 |
| Str. | 27.2 | 32.4 | 14.0 | 20.5 | 15.0 |
| Mini. | 33.3 | 35.5 | 13.5 | 21.8 | 23.1 |
| +Q. | 31.5 | 35.1 | 13.7 | 21.8 | 23.1 |
| Ours | 57.0 | 38.6 | 15.4 | 31.8 | 85.5 |
| +Q. | 55.7 | 39.8 | 15.5 | 30.0 | 81.0 |

experiments (i.e., $w_{\text{last}}$). We conduct additional experiments to analyze the impact of hyperparameters on model performance. As shown in Table 12, we find the impact of the hyperparameters is generally within 1 point.

Table 10: Performance comparison of SimLayerKV and SqueezeAttention under similar compression ratio with LLaMA3-8B-Instruct on LongBench.

| | Single-Doc. QA | | | Muti.-Doc. QA | | | Summary | | | Few-shot | | | Syn. | | Code | | |
|---|---|---|---|---|---|---|---|---|---|---|---|---|---|---|---|---|---|
| | NrtvQA | Qasper | MF-en | HotpotQA | Musique | DuReader | GovReport | QMSum | MultiNews | TREC | TriviaQA | SAMSum | PCount | PRe | LCC | RB-P | **Average** |
| ***LLaMA3-8B-Instruct*** | | | | | | | | | | | | | | | | | |
| SqueezeAttention | 20.4 | 26.9 | 31.2 | 41.3 | 19.5 | 13.4 | 24.2 | 22.4 | 23.9 | 73.0 | 90.8 | 41.6 | 3.7 | 67.0 | 56.7 | 51.7 | 38.0 |
| SimLayerKV | 23.6 | 34.7 | 43.9 | 48.0 | 22.5 | 17.0 | 26.2 | 22.5 | 26.2 | 73.5 | 89.3 | 40.6 | 3.5 | 72.5 | 58.0 | 50.7 | **40.8** |

Table 11: Performance on larger models (LLaMA3-70B-Instruct), and compression ratio across different datasets in Longbench benchmark.

| | NrtvQA | VCSUM | LCC | Average |
|---|---|---|---|---|
| **Full** | 25.6 | 15.4 | 41.6 | 28.7 |
| **SimLayerKV** | 25.5 | 13.7 | 43.5 | 27.3 |
| **Compression Ratio** | 5.50× | 7.16× | 5.21× | 5.96× |

Table 12: Effect of hyperparameters on lazy layer identification using LLama3-8B-Instruct.

| | NrtvQA | VCSUM | LCC | Average |
|---|---|---|---|---|
| $w_{\text{sink}}$ | | | | |
| **2** | 23.0 | 47.1 | 25.8 | 32.0 |
| **4** | 23.6 | 48.0 | 26.2 | 32.6 |
| **8** | 22.8 | 47.8 | 24.7 | 31.8 |
| $w_{\text{recent}}$ | | | | |
| **252** | 22.6 | 48.7 | 24.2 | 31.8 |
| **508** | 23.9 | 48.1 | 25.0 | 32.3 |
| **1020** | 23.6 | 48.0 | 26.2 | 32.6 |
| **2044** | 22.8 | 49.8 | 23.8 | 32.1 |
| $w_{\text{last}}$ | | | | |
| 16 | 22.3 | 47.0 | 25.6 | 31.6 |
| 32 | 23.6 | 48.0 | 26.2 | 32.6 |
| 64 | 24.0 | 49.3 | 24.7 | 32.7 |

### A.4 Examples about layer behavior across tokens

Additional examples of layer behavior across tokens for a given input can be found in Figure 7. The examples are randomly chosen from LongBench benchmarks. The analysis is conducted using LLama3-8B-Instruct.

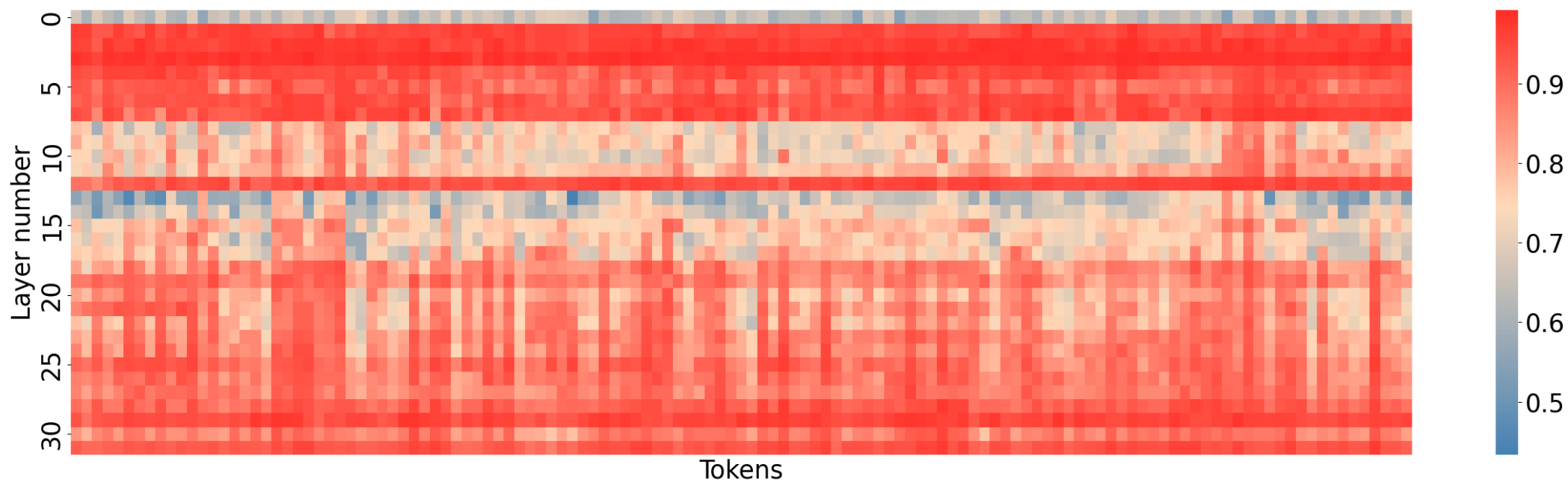

(a) Example 0

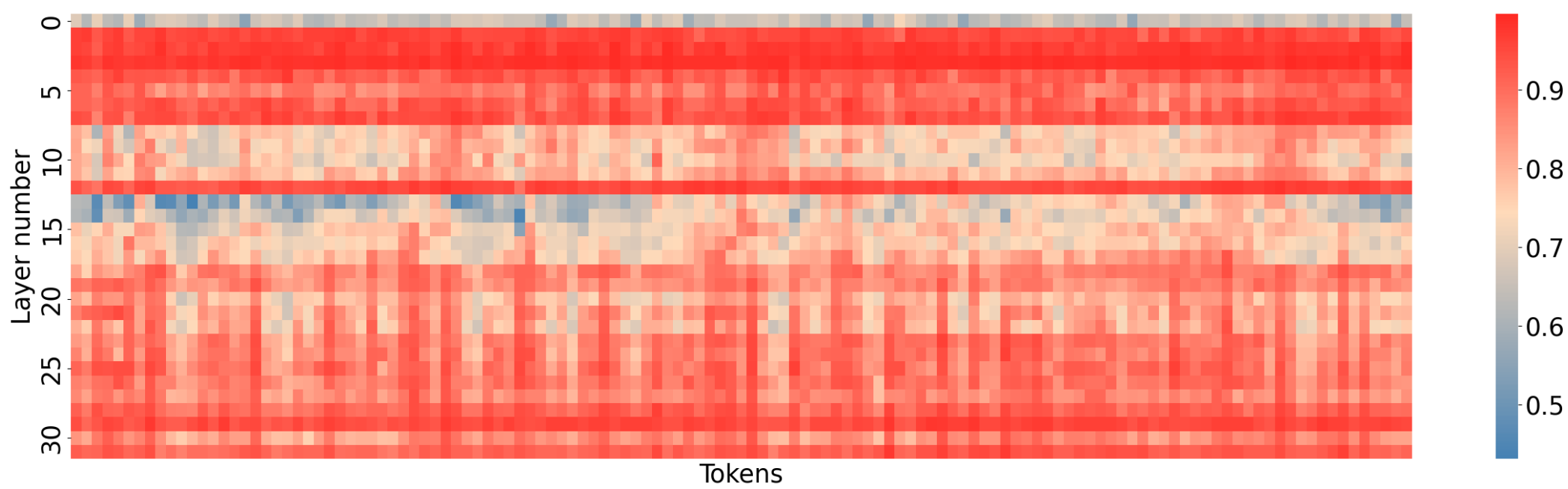

(b) Example 1

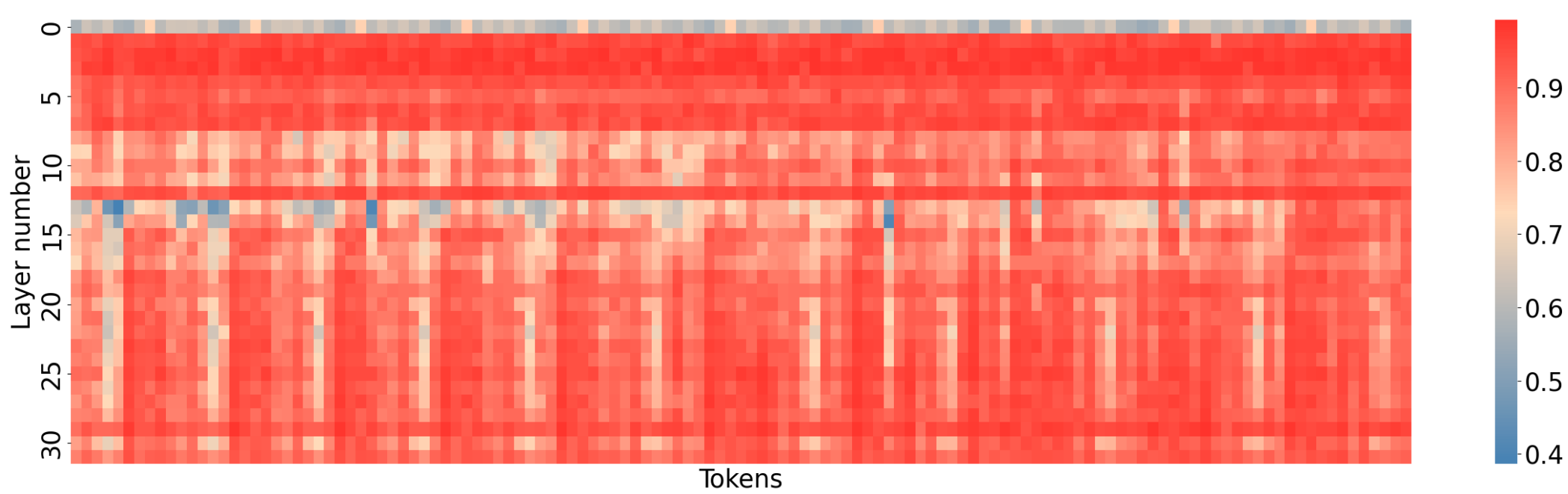

(c) Example 2

Figure 7: Additional examples about layer behavior across tokens.