# OpenReview forum: "SimLayerKV: A Simple Framework for Layer-Level KV Cache Reduction"
_ICLR.cc/2025/Conference — Submitted to ICLR 2025_

### Official Review · Reviewer_UyjG · 2024-11-02

**Soundness:** 3
**Presentation:** 3
**Contribution:** 2
**Rating:** 5
**Confidence:** 5

**Summary:**

The paper introduces SimLayerKV, a method aimed at addressing the increased memory requirements for storing key-value (KV) caches in large language models (LLMs) that handle long contexts. It identifies "non-lazy" layers in these LLMs that contribute significantly to modeling long-range dependencies. By implementing a novel KV cache eviction strategy that selectively drops caches in less critical layers based on attention weight patterns, SimLayerKV effectively reduces memory usage related to inter-layer KV cache redundancies.

**Strengths:**

- The study identifies non-lazy layers within large language models and introduces an innovative KV cache eviction method that significantly reduces the memory usage of KV caches.
- The proposed approach is training-free, generalizable, and can be implemented in just seven lines of code, demonstrating its ease of application.
- Experiments conducted on three representative LLMs across 16 tasks from the LongBench benchmark illustrate that SimLayerKV, when combined with 4-bit quantization, achieves high KV cache compression ratios with only a minimal drop in performance.

**Weaknesses:**

- The paper notes that KV caches in non-lazy layers must be fully retained, which does not fundamentally solve the substantial overhead caused by KV caches. This could still result in high memory usage. Methods like H2O are able to drastically reduce the memory footprint of KV caches.
- There is a lack of comparison with existing advanced KV cache eviction methods, such as H2O, SnapKV, and PyramidKV.
- The method proposed by the paper appears trivial, and it is unclear whether non-lazy layers will still be present in larger models or how this phenomenon may relate to different datasets.

**Questions:**

- What is the underlying cause of non-lazy layers?
- Many papers have analyzed the relationships between cross-attention layers. How does this relate to the non-lazy layers identified in this paper?

---

> ### Author Response · Authors · 2024-11-20
> **Rebuttal by Authors [1/2]**
>
> Thank you for your valuable feedback and questions. Below, we respond to the comments in Weaknesses (**W**) and Questions (**Q**).
>
> ---
> **W1: SimLayerKV can not drastically reduce the memory footprint of KV caches like H2O.**
>
> SimLayerKV is specifically designed to address inter-layer KV cache redundancies, which distinguishes it from methods targeting intra-layer redundancies. While intra-layer methods like H2O and SnapKV focus on selecting important tokens within individual layers to reduce redundancy, inter-layer methods like SimLayerKV operate on a more coarse-grained, layer-wise basis to minimize redundancy across different layers. Common inter-layer approaches include local-global attention [$\\textrm{\\color{blue}A}$] and cross-layer attention [$\\textrm{\\color{blue}B, C}$], both of which are widely adopted in current LLMs. MiniCache serves as a tuning-free implementation of cross-layer attention, while our SimLayerKV provides a tuning-free method for applying local-global attention.
>
> To illustrate the orthogonality between inter-layer and intra-layer KV cache compression methods, we select SnapKV, a cutting-edge method for intra-layer KV cache reduction, as a representative baseline. Following your suggestions, we conduct additional experiments combining SimLayerKV with SnapKV. In these experiments, SnapKV is applied to compress the KV cache for non-lazy layers, while SimLayerKV operations are retained for lazy layers. To maintain consistency with $\\textrm{\\color{red}Table 7}$, we use Qwen2.5-3B-chat-32K in this analysis. The compression ratio of SnapKV is attributed to the GQA mechanism. A detailed explanation is provided in $\\textrm{\\color{red}Appendix A.3}$.
>
> ||CompressionRatio  |Average| NrtvQA | Qasper |MFen|MFzh|HotpotQA|2WikiMQA|Musique|DuReader|
> | -------- | -------- | -------- | -------- | -------- | -------- |-------- |-------- |-------- |-------- |-------- |
> | SnapKV    |1.2$\times$|36.5| 21.6     | 32.9     |42.4     |49.9     |40.5    |38.7     |16.0    |24.1   |
> |SimLayerKV + SnapKV    |**1.7**$\times$|**37.6**    | 20.2    | 32.3     |43.0     |50.0     |48.8     |37.6     |20.7     |22.7     |
>
> ||GovR.|QMSum|MultiN.|VCSUM|TREC|TriviaQA|SAMSum|LSHT|PC.|PRe|PRz|LCC|RBP|
> |--------|-------- |-------- |-------- |-------- |-------- |-------- |-------- |-------- |-------- |-------- |-------- |-------- |-------- |
> |SnapKV   |22.0    |23.0     |22.5     |13.2     |63.0     |88.1     |43.5     |34.0     |3.5|45.0|34.3|55.1|53.9|
> |SimLayerKV + SnapKV   |21.9    |22.8     |22.4     |12.8    |65.5     |87.8    |43.4    |39.0    |4.0    |43.0  |41.0      |57.4     |53.8     |
>
>
> As shown in the table, our SimLayerKV can be combined with the intra-layer KV cache compression method to further reduce the KV cache while maintaining performance. This suggests that SimLayerKV is orthogonal to existing methods that focus on reducing intra-layer KV cache redundancies.
>
> ---
>
>
> **W2: Lack of comparison with H2O, SnapKV, and PyramidKV.**
>
> The original SnapKV paper demonstrates that it outperforms H2O in terms of efficiency and effectiveness. Therefore, we focused our comparisons on SnapKV in $\\textrm{\\color{red}Table 7}$ to provide a benchmark against the leading existing intra-layer KV cache eviction method. As shown in the table, SimLayerKV achieves performance comparable to SnapKV but offers a slightly higher compression ratio, with SimLayerKV at 1.75$\times$ and SnapKV at 1.45$\times$ on average, and the two methods are entirely orthogonal. Regarding PyramidKV, we implemented its pyramid-based KV cache retention strategy and included experimental results in $\\textrm{\\color{red}Figure 6}$ of our paper. Our findings showed that PyramidKV did not perform as well as our proposed SimLayerKV.
>
> ---
>
> **W3: Lazy layers in larger models \& different datasets.**
>
> Following your suggestion, we conducted additional experiments with LLaMA3-70B-Instruct and evaluated the compression ratio and corresponding performance of our SimLayerKV (w/o quantization) in tasks from the LongBench benchmark. We limited our report to these three different tasks (QA, summarization, and code) due to time constraints.  The results show that lazy layers are more noticeably present in larger models, and our method successfully compresses KV caches while maintaining performance. Furthermore, we observed that the phenomenon of lazy layers is consistent across different datasets. We are still working on obtaining results on additional tasks and LLaMA-400B.
>
> |  | NrtvQA | VCSUM|LCC|Average|
> | -------- | -------- | -------- | -------- | -------- |
> |  Full  | 25.6     | 15.4     |41.6     |28.7     |
> |  SimLayerKV  | 25.5| 13.7    |43.5    |27.3     |
> |  Compression ratio | 5.50$\times$    |7.16$\times$    |5.21$\times$    |5.96$\times$       |

---

> ### Author Response · Authors · 2024-11-20
> **Rebuttal by Authors [2/2]**
>
> **Q1: The underlying cause of non-lazy layers.**
>
> As we mentioned in $\\textrm{\\color{red}lines 197-203}$ of our paper, the lazy layers are also referred to as ''attention sinks''. We choose to describe it as ''lazy behavior'' to highlight shortcuts in the attention mechanism. Attention sinks happen when attending to all tokens is unnecessary, the model redirects redundant attention to sink tokens [$\\textrm{\\color{blue}D,E}$]. Previous research also found that not every attention layer contributes equally to the model's performance [$\\textrm{\\color{blue}F,G}$]. Non-lazy layers exist because the model still requires certain layers to attend to all tokens to capture long-range dependencies within the input.
>
> ---
>
> **Q2: Relationships between cross-attention layers and non-lazy layers.**
>
> We are confused about the term ''cross-attention layers'', and we assume you are referring to ''cross-layer attention''[$\\textrm{\\color{blue}B}$]. In our analysis of models like LLaMA3, we did not observe clear evidence of cross-layer attention patterns. Instead, some layers exhibit attention patterns similar to local-global attention [$\\textrm{\\color{blue}A}$]. This led us to propose SimLayerKV, leveraging these patterns to improve KV cache compression without tuning. In fact, MiniCache can be seen as a tuning-free implementation of cross-layer attention. Our experiments demonstrate that SimLayerKV outperforms MiniCache, achieving 4.8% better performance on average on LongBench with higher compression ratios.
>
> We hope this clarifies your question. If we've misunderstood, please let us know.
>
> ---
>
> [A] Gemma Team. Gemma 2: Improving open language models at a practical size.
>
> [B] William Brandon et al. Reducing Transformer Key-Value Cache Size with Cross-Layer Attention. NeurIPS 2024.
>
> [C] Tencent Hunyuan Team. Hunyuan-Large: An Open-Source MoE Model with 52 Billion Activated Parameters by Tencent.
>
> [D] Guangxuan Xiao et al. Efficient streaming language models with attention sinks. ICLR 2024.
>
> [E] Mingjie Sun et al. Massive activations in large language models. COLM 2024.
>
> [F] Shwai He et al. What Matters in Transformers? Not All Attention is Needed.
>
> [G] Suyu Ge et al. A little goes a long way: Efficient long context training and inference with partial contexts.

---

> ### Author Response · Authors · 2024-11-25
> **Summary of our Rebuttal**
>
> Dear Reviewer UyjG,
>
> Thank you again for your valuable feedback. We would like to kindly remind you that we have included the following updates:
>
> - **`Results in LLaMA 70B`** (in 1/2): Performance - $1.4$% compared with Full, with a compression ratio of $5.96 \times$.
>
> - **`Orthogonal with SnapKV`** (in 1/2): Performance + $1.1$%, with a compression ratio from $1.2 \times$ (SnapKV)  to $1.7 \times$ (SnapKV+SimLayerKV).
>
> - **`Comparison with SnapKV`** (in 1/2): Results in Appendix Table 7.
>
> - **`Additional`**: Explanations addressing other weaknesses and questions.
>
> As the discussion period is nearing its end in two days, we look forward to hearing whether our responses have addressed your concerns. We would be happy to address any additional comments or questions you may have.
>
> Best,
>
> The Authors

---

> ### Author Response · Authors · 2024-12-03
> **Final Request for Discussion: Your Feedback Is Invaluable**
>
> Dear Reviewer UyjG,
>
> Thank you again for your valuable comments and suggestions. As the **final day** of the extended reviewing period approaches, we wanted to kindly follow up on the responses we provided to your thoughtful reviews.
>
> Within the first week, we carefully addressed each of your questions to the best of our ability. Over the past two weeks, we have been eagerly awaiting your feedback but have not yet received any further discussion.
>
> We sincerely look forward to any additional comments or concerns you may have and will do our best to address them promptly within the remaining time.
>
> Thank you again for your consideration. We understand how busy this period can be, and your dedication to the review process means a great deal.
>
> Best regards,  \
> The Authors

---

### Official Review · Reviewer_fSab · 2024-11-03

**Soundness:** 2
**Presentation:** 3
**Contribution:** 2
**Rating:** 6
**Confidence:** 4

**Summary:**

This paper proposes a simple and effective method, SimLayerKV, which reduces inter-layer KV cache redundancy by selectively discarding the cache of “lazy” layers. The authors found that, in long-context LLMs, certain layers contribute minimally to modeling long-distance dependencies, displaying “lazy” behavior. By analyzing attention weight patterns, they observed that these lazy layers consistently exhibit this behavior throughout the generation process for a given input. SimLayerKV identifies these lazy layers and reduces their KV cache accordingly without altering the cache of non-lazy layers or merging caches across layers. Extensive experiments on three representative LLMs demonstrate that SimLayerKV, combined with 4-bit quantization, achieves a 5x KV cache compression rate with only a 1.2% performance drop.

**Strengths:**

1. This paper addresses a good research topic: efficient LLM inference.

2. The paper is well-organized.

3. The proposed method is clearly presented.

**Weaknesses:**

1. The designed identification algorithm does not meets the observation. It will also treat the layer that attend all of the tokens, including the intial, the recent tokens, and other tokens in the sequence as the lazy layer.

2. Why only use the only one token rather than few tokens, as in prefilling stage, in decoding to detect the lazy layers?

3. It seems that there are a lot of parameters need to be manually set in this algorithm, including the X_intial, X_recent, and W_last, making the designed algorithm less practical. Moreover, these parameters are also not covered in the ablation study and the authors did not explain how they configure them in the experiments.

**Questions:**

See above.

---

> ### Author Response · Authors · 2024-11-20
> **Rebuttal by Authors**
>
> Thank you for your valuable feedback and questions. Below, we respond to the comments in Weaknesses (**W**) and Questions (**Q**).
>
> ---
> **W1: The designed identification algorithm does not meet the observation.**
>
> We want to clarify that our algorithm does not classify layers that attend to all tokens as lazy layers. If the attention is not predominately focused on sink tokens and recent tokens, we will not perform any operation on that layer. The experimental results in $\\textrm{\\color{red}Figure 6}$ further demonstrate that using our identification method achieves better results compared to other strategies.
>
> ---
>
> **W2: Why only use one token in decoding to detect the lazy layers?**
>
> We use only the first token in decoding to detect lazy layers because it proves to be more efficient without reducing effectiveness. In our preliminary experiments, employing multiple tokens did not yield significant performance enhancements. Furthermore, by identifying and pruning lazy layers at the very start of decoding, we enhance the generation speed of subsequent tokens by minimizing the I/O overhead linked to the KV cache.
>
> ---
>
> **W3: Ablation study about hyper-parameters.**
>
> We adopt these hyperparameters either directly from StreamingLLM[$\\textrm{\\color{blue}A}$] (i.e., $w_\text{sink}$ and $w_\text{recent}$), ensuring consistency with established practices in the field, or through preliminary experiments (i.e., $w_\text{last}$, and $\delta$). In response to your suggestion, we also conducted additional experiments to analyze the impact of other hyperparameters ($w_\text{sink}$, $w_\text{recent}$, and $w_\text{last}$) on model performance. We found that the impact of the hyperparameters is generally within 1 point.
>
>
> |$w_\text{sink}$ | 2 |4 |8 |
> | -------- | -------- | -------- |-------- |
> |NrtvQA    |   23.0   |  23.6   |22.8    |
> |HotpotQA| 47.1    |48.0     |47.8     |
> |MuSiQue   | 25.8     |26.2    |24.7   |
> |Avg.|32.0|32.6|31.8|
>
> |$w_\text{recent}$ | 252 | 508 |1020 |2044 |
> | -------- | -------- | -------- |-------- |-------- |
> |NrtvQA    | 22.6     | 23.9     |23.6     |22.8    |
> |HotpotQA| 48.7     | 48.1     |48.0     |49.8   |
> |MuSiQue  | 24.2     | 25.0     |26.2     |23.8    |
> |Avg.|31.8|32.3|32.6|32.1|
>
> |$w_\text{last}$| 16 |32 |64 |
> | -------- | -------- | -------- |-------- |
> |NrtvQA   | 22.3     |23.6     |24.0    |
> |HotpotQA| 47.0     |48.0     |49.3     |
> |MuSiQue  | 25.6     |26.2     |24.7    |
> |Avg.|31.6|32.6|32.7|
>
>
> The threshold $\delta$ serves as a KV cache budget and can be chosen based on the desired compression ratio. Once $\delta$ is defined using a small subset of data for a given model, it remains unchanged across different tasks, making it practical and easy to apply. The impact of threshold $\delta$ can be found in $\\textrm{\\color{red}Figure 5}$.
>
> ---
>
> [A] Guangxuan Xiao et al. Efficient streaming language models with attention sinks. ICLR 2024.

---

> > ### Comment · Reviewer_fSab · 2024-11-23
> >
> > Thanks for the clarification. I have raised my score to 6. Good Luck!

---

> > > ### Author Response · Authors · 2024-11-24
> > > **Thank you for your support and raising the score**
> > >
> > > We greatly appreciate your valuable feedback and the score improvement. We will further polish the paper in the final revision. Thank you!

---

### Official Review · Reviewer_YxRN · 2024-11-03

**Soundness:** 2
**Presentation:** 2
**Contribution:** 2
**Rating:** 5
**Confidence:** 5

**Summary:**

This paper aims to optimize the efficiency of the LLM inference by reducing the KV embeddings that are needed to be cached in the memory. Based on the observations of the attention weights of different layers, the authors propose an algorithm to identify the “lazy” layers, whose attentions weights of the initial tokens and recent tokens are predominately larger than those of other tokens, and then trim the KV cache of these lazy layers so as to reduce the memory cost. Some experiments on LLaMA2-7B, LLaMA-3-8B, and Mistral-7B show the effectiveness of the proposed method.

**Strengths:**

1. The research question is interesting and promising for the highly-efficient inference of LLM. Since the KV cache grows linearly with the number of layers, it’s natural to optimize this problem from the layer’s perspective. Recently, increasingly more studies are focused on this topic.
2. From what I understand, the proposed method might be able to save the memory cost not only for decoding, but also prefilling, which is useful for those case where the prompt is way longer than the generated response.
3. The paper has reasonable design of experiments.

**Weaknesses:**

1. The way to identify the lazy layers involves quite a few hyper-parameters, such as the length of $X_{last}$, $X_{initial}$, $X_{recent}$, and $\delta$. There is no systematic way to tune those hyper-parameters, which makes it too empirical.
2. Since there are two ways to identify the lazy layers, does it mean the lazy layers in prefilling and decoding are different?
3. Can we identify the lazy layer on-the-fly during the inference? because in practice we usually do not have a small subset of data to identify the lazy layers first, then conduct the real inference.
4. A big missing part of this paper is the result of efficiency, like token throughput, latency or memory cost of the proposed method, since the goal of the algorithm is to optimize the efficiency.
5. There are quite a few related works trying to identify the optimal KV cache strategies for different layers [1][2][3], it’s unclear what are the cons and pros of those methods compared with the proposed one.

[1] https://arxiv.org/pdf/2410.10819

[2] https://arxiv.org/pdf/2404.04793

[3] https://arxiv.org/pdf/2310.01801

**Questions:**

please refer to the Weaknesses

---

> ### Author Response · Authors · 2024-11-20
> **Rebuttal by Authors [1/2]**
>
> Thank you for your valuable feedback and questions. Below, we respond to the comments in Weaknesses (**W**) and Questions (**Q**).
>
> ---
> **W1: Systematic way to tune hyper-parameters.**
>
> We adopt these hyperparameters either directly from StreamingLLM[$\\textrm{\\color{blue}A}$] (i.e., $w_\text{sink}$ and $w_\text{recent}$), ensuring consistency with established practices in the field, or through preliminary experiments (i.e., $w_\text{last}$, and $\delta$). In response to your suggestion, we also conducted additional experiments to analyze the impact of other hyperparameters ($w_\text{sink}$, $w_\text{recent}$, and $w_\text{last}$) on model performance. We found that the impact of the hyperparameters is generally within 1 point.
>
>
> |$w_\text{sink}$ | 2 |4 |8 |
> | -------- | -------- | -------- |-------- |
> |NrtvQA    |   23.0   |  23.6   |22.8    |
> |HotpotQA| 47.1    |48.0     |47.8     |
> |MuSiQue   | 25.8     |26.2    |24.7   |
> |Avg.|32.0|32.6|31.8|
>
> |$w_\text{recent}$ | 252 | 508 |1020 |2044 |
> | -------- | -------- | -------- |-------- |-------- |
> |NrtvQA    | 22.6     | 23.9     |23.6     |22.8    |
> |HotpotQA| 48.7     | 48.1     |48.0     |49.8   |
> |MuSiQue  | 24.2     | 25.0     |26.2     |23.8    |
> |Avg.|31.8|32.3|32.6|32.1|
>
> |$w_\text{last}$| 16 |32 |64 |
> | -------- | -------- | -------- |-------- |
> |NrtvQA   | 22.3     |23.6     |24.0    |
> |HotpotQA| 47.0     |48.0     |49.3     |
> |MuSiQue  | 25.6     |26.2     |24.7    |
> |Avg.|31.6|32.6|32.7|
>
> The threshold $\delta$ serves as a KV cache budget and can be chosen based on the desired compression ratio. Once $\delta$ is defined using a small subset of data for a given model, it remains unchanged across different tasks, making it practical and easy to apply. The impact of threshold $\delta$ can be found in $\\textrm{\\color{red}Figure 5}$.
>
> ---
>
> **W2: Are lazy layers in prefilling and decoding different?**
>
> Our experiments show a high overlap between lazy layers identified in the prefilling and decoding stages, indicating they are largely consistent. As shown in $\\textrm{\\color{red}Figure 6}$, the performance of both strategies is comparable. Identifying lazy layers during the prefilling stage reduces peak memory usage and requires only an extra $w_{\text{last}} \times w_{\text{recent}}$ matrix multiplication if used with FlashAttention. The additional time cost is between 0.0058 (4K sequence length) and 0.0014 (32K sequence length) of the full prefilling time. In contrast, identifying lazy layers during decoding does not reduce peak memory usage but leverages readily available attention weights. Users can choose the strategy based on their specific needs.
>
> ---
>
> **W3: Identify lazy layers in an on-the-fly manner.**
>
> We would like to clarify that in our SimLayerKV method, the identification of lazy layers is indeed performed on-the-fly during inference. This means that we do not require a preliminary subset of data to identify lazy layers before conducting the actual inference. As mentioned in $\\textrm{\\color{red}lines 270-273}$ of our paper, our method dynamically determines lazy layers based on only the current input data during the inference process.

---

> ### Author Response · Authors · 2024-11-20
> **Rebuttal by Authors [2/2]**
>
> **W4: The result of efficiency, like token throughput.**
>
>  We have included the memory cost of the proposed method in $\\textrm{\\color{red}Table 2}$ in our paper. Following your suggestion, we have now included the throughput results as well. Using LLaMA3-8B on the Ruler benchmark, we measured throughput under the maximum batch size for input sequence lengths of 4K, 8K, 16K, and 32K on A100. The throughput (tokens/s) for SimLayerKV relative to the Full method was **1.44×**, **1.78×**, **2.17×**, and **1.75×**, respectively.
>
> ---
>
> **W5: Comparision with related works [1][2][3].**
>
> We primarily compared our method with MiniCache, as it is the current state-of-the-art in inter-layer KV cache compression. Regarding Ref [1], it was submitted to arXiv after the ICLR 2025 submission deadline, and thus was not included in our original submission. Additionally, Ref [1] involves continue training, while our SimLayerKV is a tuning-free, plug-and-play method that can be applied without training the model. This makes SimLayerKV more practical for immediate deployment.
>
> For Ref [3], we attempted to test it, but the source code repository is still empty as of now. Therefore, we had to reimplement it based on the paper. However, this reimplementation requires instantiating the attention score matrix (batch_size * num_heads * seq_len * seq_len) as described in Equation 1 of Ref [3]. In our experiments, this leads to OOM issues. Ref [1] also found the OOM problem of Ref [3] in LongBench.  In contrast, our SimLayerKV is compatible with FlashAttention and requires only an extra $w_{\text{last}} \cdot w_{\text{recent}}$ matrix multiplication operation, making it more efficient and scalable for long-context applications.
>
> In addition, both Ref [1] and [3] are head-level KV cache compression. When running LLMs, tensor parallelism (TP) is widely used in the industry to distribute computation across multiple GPUs. Typical, A layer consists of multiple KV heads (e.g., 8 heads per layer), with each head assigned to a separate GPU. If different heads within the same layer have varying sizes of compressed KV caches, GPUs handling smaller caches would have to wait for those with larger caches to complete their computations. This synchronization bottleneck can eliminate any potential latency reductions achieved through compression. Considering these factors, we do not consider them as our baselines.
>
> For Ref [2], thanks for pointing out this related work. Following your suggestions, we added the comparison of Ref [2] in LongBench benchmark using LLaMA3-8B-Instruct. The results indicate that SimLayerKV preserves long-context capabilities better than Ref [2] under similar compression ratios. Additionally, Ref [2] can not reduce peak memory usage during prefilling.
>
> |  | Average|NrtvQA | Qasper |MF-en|HotpotQA|Musique|DuReader|GovReport|
> | -------- | -------- | -------- | -------- | -------- | -------- |-------- |-------- |-------- |
> | SqueezeAttention   |38.0| 20.4     | 26.9     |31.2     |41.3     | 19.5    | 13.4   |24.2     |
> |SimLayerKV    | **40.8**     |23.6    | 34.7    |43.9     |48.0    |22.5     |17.0    |26.2    |
>
> |  |QMSum|MultiNews|TREC|TriviaQA|SAMSum|PCount|PRe|LCC|RB-P|
> |--------|-------- |-------- |-------- |-------- |-------- |-------- |-------- |-------- |-------- |
> | SqueezeAttention|22.4     |23.9     |73.0    |90.8     |41.6     |3.7     |67.0     |56.7  |51.7    |
> |SimLayerKV   |22.5    |26.2   |73.5    |89.3    |40.6   |3.5     |72.5     |58.0     |50.7    |
>
> ---
>
> [A] Guangxuan Xiao et al. Efficient streaming language models with attention sinks. ICLR 2024.

---

> ### Author Response · Authors · 2024-11-25
> **Summary of our Rebuttal**
>
> Dear Reviewer YxRN,
>
> Thank you again for your valuable feedback. We would like to kindly remind you that we have included the following updates:
>
> - **`Hyperparameter robustness`** (in 1/2): Fluctuates within 1 point.
>
> - **`Clarification about identification`** (in 1/2): We identify lazy layers on-the-fly without the need for a small dataset.
>
> - **`Throughput improvement`** (in 2/2): $1.44\times$ (4K), $1.78\times$ (8K), $2.17\times$ (16K), and $1.75\times$ (32K).
>
> - **`Comparison with SqueezeAttention`** (in 2/2): Performance + $2.8$% under same compression ratio.
>
> - **`Additional`**: Explanations addressing other weaknesses and questions.
>
> As the discussion period is nearing its end in two days, we look forward to hearing whether our responses have addressed your concerns. We would be happy to address any additional comments or questions you may have.
>
> Best,
>
> The Authors

---

> > ### Comment · Reviewer_YxRN · 2024-12-02
> > **Thanks for your response**
> >
> > Dear authors,
> >
> > Thanks for your response. Since the efficiency is the motivation of compression, I would suggest a more comprehensive comparison of throughput or latency with other SOTA works, instead of the Full cache baseline.
> > I'd like to increase my score to 5 for the current version.

---

> ### Author Response · Authors · 2024-12-02
> **Thank you for your support and raising the score**
>
> Thank you for your valuable feedback and for raising the score! Following your suggestions, we provide a comparison of throughput with SOTA works in inter-layer KV cache compression (i.e., SqueezeAttention and Minicache) on the Ruler benchmark.
>
> |Throughput | 4K |8K |16K |32K |
> | -------- | -------- | -------- |-------- |-------- |
> |SqueezeAttention| 1.03$\times$|1.09$\times$|1.12$\times$|1.04$\times$|
> |MiniCache   |   1.26$\times$   |  1.29$\times$   |1.52$\times$   |1.41$\times$|
> |SimLayerKV (ours)   | 1.44$\times$     |1.78$\times$ |2.17$\times$   |1.75$\times$|
>
> As shown in the table, our SimLayerKV achieves the highest throughput.
> For SqueezeAttention, under a similar compression ratio, the low throughput is attributed to its inability to reduce peak memory usage during prefilling, as it requires completing the prefilling of all layers before performing compression. This limitation prevents the use of larger batch sizes, meaning it can only improve throughput by reducing the I/O in attention.
> For Minicache, the lower throughput is due to its compression ratio being smaller than that of our SimLayerKV (the compression ratio is $1.26\times$ for MiniCache and $1.64\times$ for SimLayerKV, as detailed in $\\textrm{\\color{red}Table 2}$).
>
> We will incorporate these comparisons into the final revision and further polish the paper. Thank you again for your valuable insights and support!

---

### Official Review · Reviewer_KtMT · 2024-11-07

**Soundness:** 3
**Presentation:** 3
**Contribution:** 2
**Rating:** 5
**Confidence:** 4

**Summary:**

With the recent increase in context lengths of large language models (LLMs), the memory required to store key-value (KV) cache has grown significantly. This paper aims to reduce inter-layer KV cache redundancies by selectively dropping caches in identified "lazy" layers. The paper first observes that certain layers in long-context LLMs are "lazy," primarily focusing on semantically unimportant tokens (the initial few tokens and the most recent few tokens) when performing attention. Furthermore, lazy layers are less important than non-lazy layers in long-context generation. The paper also finds that the laziness behavior is consistent across tokens for a given input and is easily identifiable. Based on these observations, the paper proposes SimLayerKV, a simple strategy that identifies lazy layers at either the prefill or decode stage and trims the lazy layers to reduce inter-layer KV cache redundancy. To demonstrate SimLayerKV's effectiveness, the paper evaluates it on popular benchmarks like LongBench and Ruler, showing a maximum compression ratio of 5x with only a 1.2% drop in performance. Unlike existing works, this paper is distinct in leveraging inter-layer KV cache redundancies and requires no additional training.

**Strengths:**

- The paper is novel in its exploration of better inter-layer KV cache trimming without additional training.
- The paper is well written

**Weaknesses:**

- It is not clear why SimLayerKV is orthogonal to existing KV cache trimming or compressing methods
- The compression ratio of 1.6x on average without 4 bit Quantization is not significant
- There is performance degradation on more complex tasks
- The proposed way of identifying lazy layers is not flexible enough

**Questions:**

- In Figure 2, the paper demonstrates the attention patterns during long-context generation in layers 0, 10, 20, and 30, thereby categorizing layers into two types: lazy and non-lazy. How do the insights gained regarding attention patterns in this paper compare to prior work, such as MInference1.0, which identifies three sparse patterns (A-shape, Vertical-Slash, and Block-Sparse)? Do these findings align?

- Could you clarify the lazy layer identification algorithm further? The paper suggests two methods for identifying lazy layers—during the prefill stage and the decode stage. Specifically, when is this identification executed during online inference? How frequently is it updated during benchmarking, and how often would it be updated in an online inference scenario? For multi-round conversations, where inputs from a single user may vary significantly between rounds, how does SimLayerKV address this in its design?

- In Section 2 on related work, the paper references prior research on KV cache trimming, compression, and selection. It includes comparisons with MiniCache, StreamingLLM, and SnapKV as baselines. Why does the paper not include comparisons with more intra-layer trimming methods? Can you provide examples illustrating why inter-layer methods are orthogonal to intra-layer methods? Additionally, have you conducted experiments to demonstrate that integrating these two approaches does not significantly degrade performance?

- In Table 1, the paper compares the performance of SimLayerKV and baseline methods. On the LongBench benchmarks, why does SimLayerKV+Q outperform SimLayerKV in many cases, especially in nearly half of the tests for the Mistral-7B-Instruct model?

- What configuration is used for the StreamingLLM baseline? In Table 6, StreamingLLM achieves a 6-8x higher compression ratio than SimLayerKV; could you provide additional comparison results where both methods achieve similar compression ratios?

- In Section 6.3 on Ruler experiments, SimLayerKV shows a performance drop (8.2% on average) on Multiple Queries NIAH and significant degradation in Common Words Extraction (from 75.1% to 48.6% for a 32k context length). The paper attributes this to the data-dependent nature of lazy layer identification, specifically a fixed selection of lazy layers across the entire benchmark task. Have you conducted experiments to verify this hypothesis? Additionally, have you considered dynamically updating the selected lazy layers during runtime? What are possible solutions to fix this issue?

- What is the overhead introduced by lazy layer identification in terms of latency, and how does it affect the system’s overall throughput? Have you conducted end-to-end serving experiments to demonstrate SimLayerKV's deployment potential?

- How might performance and compression ratios change if individual heads within a layer are considered? Have additional experiments been conducted on this aspect? Would controlling for smaller granularities potentially improve performance, or could it lead to worse outcomes?

---

> ### Author Response · Authors · 2024-11-20
> **Rebuttal by Authors [1/5]**
>
> Thank you for your valuable feedback and questions. Below, we respond to the comments in Weaknesses (**W**) and Questions (**Q**).
>
>
> ---
>
> **W1: Why SimLayerKV is orthogonal to existing KV cache trimming methods.**
>
> SimLayerKV is specifically designed to address inter-layer KV cache redundancies, which distinguishes it from methods targeting intra-layer redundancies. While intra-layer methods focus on selecting important tokens within individual layers to reduce redundancy, inter-layer methods our like SimLayerKV operate on a more coarse-grained, layer-wise basis to minimize redundancy across different layers. Common inter-layer approaches include local-global attention [$\\textrm{\\color{blue}A}$] and cross-layer attention [$\\textrm{\\color{blue}B, C}$], both of which are widely adopted in current LLMs. MiniCache serves as a tuning-free implementation of cross-layer attention, while our SimLayerKV provides a tuning-free method for applying local-global attention.
>
> To illustrate the orthogonality between inter-layer and intra-layer KV cache compression methods, we select SnapKV, a cutting-edge method for intra-layer KV cache reduction, as a representative baseline. Following your suggestions, we conduct additional experiments combining SimLayerKV with SnapKV. In these experiments, SnapKV is applied to compress the KV cache for non-lazy layers, while SimLayerKV operations are retained for lazy layers. To maintain consistency with Table 6, we use Qwen2.5-3B-chat-32K in this analysis. The compression ratio of SnapKV is attributed to the GQA mechanism. A detailed explanation is provided in $\\textrm{\\color{red}Appendix A.3}$.
>
> ||CompressionRatio  |Average| NrtvQA | Qasper |MFen|MFzh|HotpotQA|2WikiMQA|Musique|DuReader|
> | -------- | -------- | -------- | -------- | -------- | -------- |-------- |-------- |-------- |-------- |-------- |
> | SnapKV    |1.2$\times$|36.5| 21.6     | 32.9     |42.4     |49.9     |40.5    |38.7     |16.0    |24.1   |
> |SimLayerKV + SnapKV    |**1.7**$\times$|**37.6**    | 20.2    | 32.3     |43.0     |50.0     |48.8     |37.6     |20.7     |22.7     |
>
> ||GovR.|QMSum|MultiN.|VCSUM|TREC|TriviaQA|SAMSum|LSHT|PC.|PRe|PRz|LCC|RBP|
> |--------|-------- |-------- |-------- |-------- |-------- |-------- |-------- |-------- |-------- |-------- |-------- |-------- |-------- |
> |SnapKV   |22.0    |23.0     |22.5     |13.2     |63.0     |88.1     |43.5     |34.0     |3.5|45.0|34.3|55.1|53.9|
> |SimLayerKV + SnapKV   |21.9    |22.8     |22.4     |12.8    |65.5     |87.8    |43.4    |39.0    |4.0    |43.0  |41.0      |57.4     |53.8     |
>
> As shown in the table, our SimLayerKV can be combined with intra-layer KV cache compression method to further reduce the KV cache while maintaining performance. This suggests that SimLayerKV is orthogonal to existing methods that focus on reducing intra-layer KV cache redundancies.
>
> ---
>
> **W2: The compression ratio of $1.6 \times$ w/o Quantization is not significant.**
>
> Our SimLayerKV, is a tuning-free inter-layer KV cache compression approach. When compared to the current SoTA method in this domain, MiniCache, which achieves a compression ratio of only $1.26 \times$, our method achieves a notable improvement in compression ratio.
>
> ---
>
> **W3: There is performance degradation on more complex tasks.**
>
> We acknowledge that on certain synthetic complex tasks, such as CWE and MQ-NIAH, there is indeed a performance drop. However, we would like to highlight that SimLayerKV outperforms the SoTA layer-wise pruning method, MiniCache, under similar settings.

---

> ### Author Response · Authors · 2024-11-20
> **Rebuttal by Authors [2/5]**
>
> **W4: The proposed way of identifying lazy layers is not flexible enough.**
>
> We are confused about your comment that our identification method is not flexible enough, we assume that our method may involve complexity in hyperparameter selection, which might affect its flexibility in different scenarios. We adopt these hyperparameters either directly from StreamingLLM[$\\textrm{\\color{blue}D}$] (i.e., $w_\text{sink}$ and $w_\text{recent}$), ensuring consistency with established practices in the field, or through preliminary experiments (i.e., $w_\text{last}$, and $\delta$). In response to your suggestion, we also conducted additional experiments to analyze the impact of other hyperparameters ($w_\text{sink}$, $w_\text{recent}$, and $w_\text{last}$) on model performance. We found that the impact of the hyperparameters is generally within 1 point.
>
>
> |$w_\text{sink}$ | 2 |4 |8 |
> | -------- | -------- | -------- |-------- |
> |NrtvQA    |   23.0   |  23.6   |22.8    |
> |HotpotQA| 47.1    |48.0     |47.8     |
> |MuSiQue   | 25.8     |26.2    |24.7   |
> |Avg.|32.0|32.6|31.8|
>
> |$w_\text{recent}$ | 252 | 508 |1020 |2044 |
> | -------- | -------- | -------- |-------- |-------- |
> |NrtvQA    | 22.6     | 23.9     |23.6     |22.8    |
> |HotpotQA| 48.7     | 48.1     |48.0     |49.8   |
> |MuSiQue  | 24.2     | 25.0     |26.2     |23.8    |
> |Avg.|31.8|32.3|32.6|32.1|
>
> |$w_\text{last}$| 16 |32 |64 |
> | -------- | -------- | -------- |-------- |
> |NrtvQA   | 22.3     |23.6     |24.0    |
> |HotpotQA| 47.0     |48.0     |49.3     |
> |MuSiQue  | 25.6     |26.2     |24.7    |
> |Avg.|31.6|32.6|32.7|
>
> The threshold $\delta$ serves as a KV cache budget and can be chosen based on the desired compression ratio. Once $\delta$ is defined using a small subset of data for a given model, it remains unchanged across different tasks, making it practical and easy to apply. The impact of threshold $\delta$ can be found in $\\textrm{\\color{red}Figure 5}$ of our paper.
>
> ---
>
> **Q1: Do findings align with MInference1.0?**
>
> Both our work and MInference1.0 observe that certain layers within LLMs exhibit streaming properties, and aim to leverage this characteristic to reduce the computational burden of attention mechanisms. MInference focuses on speeding up **prefilling**, whereas our SimLayerKV aims to enhance KV cache compression during **decoding**. Additionally, the methodologies for identifying and utilizing these streaming layers differ. MInference1.0 assumes that streaming layers are static and can be predetermined. In contrast, our method considers the identification of lazy (i.e., streaming) layers to be data-dependent. We determine the role of each layer on-the-fly based on the input context. This dynamic approach allows us to adapt to varying attention patterns that occur with different inputs, potentially capturing more nuanced behaviors of the model.
>
> In our experiments with 100 random samples from the LongBench benchmark using LLaMa3-8B-Instruct, we observe that the attention patterns are influenced by the input to some extent. Here are the probabilities for certain layers being selected as ''lazy'':
> |Layer idx | 1 | 11 |21|31|
> | -------- | -------- | -------- |-------- |-------- |
> |Propability    | 61.9|  76.7   | 58.6     |78.9|
>
> This variability supports our approach of dynamically determining each layer's role based on the input context.

---

> ### Author Response · Authors · 2024-11-20
> **Rebuttal by Authors [3/5]**
>
> **Q2-1: When is this identification executed during online inference?**
>
> Our paper presents two distinct strategies for identifying lazy layers—one executed during the prefilling stage and the other during the decoding stage. Users can choose either strategy based on their specific needs and computational considerations.
>
> ---
>
> **Q2-2: How frequently is the layer identification updated?**
>
> The lazy layer identification is executed **once** per input without additional updates.
>
> ---
>
> **Q2-3: How does SimLayerKV address multi-round conversations in its design?**
>
> We acknowledge that multi-round conversations present additional considerations, and our SimLayerKV can be naturally extended to handle multi-round scenarios:  instead of discarding the KV caches of lazy layers after each round, we can offload them to CPU memory. At the beginning of each new round, we can re-identify the lazy layers based on the updated input context. This allows the model to adapt to changes in the conversation dynamically. In this offload setting, compared to methods like SnapKV, our SimLayerKV requires less CPU-GPU communication, since the KV caches of non-lazy layers remain on the GPU and do not need to be reloaded.
>
> We appreciate your insightful suggestion. Although we have not yet demonstrated this extension due to the absence of appropriate multi-round long-context datasets. We plan to explore in future work to further enhance our Method.
>
> ---
>
> **Q3: Comparison and combination with intra-layer KV cache reduction methods.**
>
> The comparison can be found in  $\\textrm{\\color{red}Appendix Table 7}$. As shown in the table, we can see that our SimLayerKV achieves comparable performance with SnapKV with a similar compression ratio. For the combination, please see our response to **W1** in Rebuttal [1/5].
>
> ---
>
> **Q4: Why does SimLayerKV+Q outperform SimLayerKV in many cases?**
>
> This performance improvement is also noted in the original KIVI paper [$\\textrm{\\color{blue}E}$]. In the LongBench benchmark, when combined with 4-bit quantization, nearly all models demonstrate comparable or better performance compared to their non-quantized counterparts in more than half of the tasks.
>
> ---
>
> **Q5: Comparison with StreamingLLM at similar compression ratios.**
>
> We adhere to the original configuration of StreamingLLM [$\\textrm{\\color{blue}D}$] in our experiments, where only the KV caches for sink and recent tokens are retained in each layer. Beyond the results presented in $\\textrm{\\color{red}Table 1}$, we conducted additional experiments to evaluate both methods under comparable compression ratios, as shown in $\\textrm{\\color{red}Figure 6 (c, e, f)}$. These results demonstrate that SimLayerKV consistently outperforms StreamingLLM, even when maintaining the same compression ratio.

---

> ### Author Response · Authors · 2024-11-20
> **Rebuttal by Authors [4/5]**
>
> **Q6: Reason and possible solution to the performance drop in MQ-NIAH and CWE.**
>
> We would like to clarify that our method does not use a fixed selection of lazy layers across the entire benchmark task. The identification of lazy layers in SimLayerKV is performed dynamically based on the input context at the beginning of the inference process. However, for tasks with inputs containing multiple queries, varying the number of queries in a single input should ideally result in different layers being identified as lazy and reduced accordingly. In our current implementation, the same layers are reduced regardless of the query count within one input, which contributes to the performance drop observed in such tasks.
>
> A promising direction to address this limitation is integrating SimLayerKV with speculative decoding frameworks to achieve lossless acceleration to better adapt to varying query requirements, such as Triforce[$\\textrm{\\color{blue}F}$].
>
> ---
>
> **Q7: Latency and throughput.**
>
> The overhead introduced is minimal if the identification happens in the decoding phase, since we need to get the attention score no matter use our method or not. If the identification occurs during the prefill phase, we leverage the log-sum-exp returned by FlashAttention to minimize any impact on throughput. We provide the source code of the identification algorithm here.
> ```python3
> attn_out, lse = flash_attn(q, k, v, causal=True, return_lse=True)
>
> # identification
> # w_last = 32, w_sink=4, w_recent=1020
> q_last = q[:, -w_last:].permute(0, 2, 1, 3)
> k_comb = torch.cat([k[:, 0:w_sink], k[:, -w_recent:]], dim=1).permute(0, 2, 3, 1)
> log_lazy_weight = torch.matmul(q_last, k_comb).logsumexp(dim=-1) - lse
> ```
> In our experiments, the reduction in throughput compared to the original (assumed to be 1) is neglectable — between **0.0058** and **0.0014**, depending on the sequence length (with longer sequences experiencing smaller reductions, in the range of 4K to 32K tokens).
>
> Following your suggestion, we have now included the throughput results as well. Using LLaMA3-8B on the Ruler benchmark, we measured throughput under the maximum batch size for input sequence lengths of 4K, 8K, 16K, and 32K. The throughput (tokens/s) for SimLayerKV relative to the Full method was 1.44×, 1.78×, 2.17×, and 1.75×, respectively.

---

> ### Author Response · Authors · 2024-11-20
> **Rebuttal by Authors [5/5]**
>
> **Q8: Compression at head level.**
>
> We understand the concerns regarding why head-level KV cache compression was not adopted. In our initial investigations, we explored head-level compression, and found it offered only marginal performance improvements over the layer-wise approach.
>
> Additionally, when running LLMs, tensor parallelism (TP) is widely used in the industry to distribute computation across multiple GPUs. Typical, A layer consists of multiple KV heads (e.g., 8 heads per layer), with each head assigned to a separate GPU. If different heads within the same layer have varying sizes of compressed KV caches, GPUs handling smaller caches would have to wait for those with larger caches to complete their computations. This synchronization bottleneck can eliminate any potential latency reductions achieved through compression.
>
> Considering these factors, we focused on layer-level compression with SimLayerKV, which strikes a balance between practicality and performance without introducing additional synchronization overhead.
>
> ---
>
> [A] Gemma Team. Gemma 2: Improving open language models at a practical size.
>
> [B] William Brandon et al. Reducing Transformer Key-Value Cache Size with Cross-Layer Attention. NeurIPS 2024.
>
> [C] Tencent Hunyuan Team. Hunyuan-Large: An Open-Source MoE Model with 52 Billion Activated Parameters by Tencent.
>
> [D] Guangxuan Xiao et al. Efficient streaming language models with attention sinks. ICLR 2024.
>
> [E] Zirui Liu et al. Kivi: A tuning-free a symmetric 2bit quantization for kvcache. ICML 2024.
>
> [F] Hanshi Sun et al. TriForce: Lossless Acceleration of Long Sequence Generation with Hierarchical Speculative Decoding. COLM 2024.

---

> ### Author Response · Authors · 2024-11-25
> **Summary of our Rebuttal**
>
> Dear Reviewer KtMT,
>
> Thank you again for your valuable feedback. We would like to kindly remind you that we have included the following updates:
>
> - **`Orthogonal with SnapKV`** (in rebuttal 1/5): Performance + $1.1$%, with a compression ratio from $1.2 \times$ (SnapKV) to $1.7 \times$ (SnapKV+SimLayerKV).
>
> - **`Throughput improvement`** (in rebuttal 4/5): $1.44\times$ (4K), $1.78\times$ (8K), $2.17\times$ (16K), and $1.75\times$ (32K).
>
> - **`Identification cost`** (in rebuttal 4/5): The cost is between $0.0058$ (4K) and $0.0014$ (32K).
>
> - **`Additional`**: Ablation studies and explanations addressing other weaknesses and questions
>
>  As the discussion period is nearing its end in two days, we look forward to hearing whether our responses have addressed your concerns. We would be happy to address any additional comments or questions you may have.
>
> Best,
>
> The Authors

---

> ### Author Response · Authors · 2024-12-03
> **Final Request for Discussion: Your Feedback Is Invaluable**
>
> Dear Reviewer KtMT,
>
> Thank you again for your valuable comments and suggestions. As the **final day** of the extended reviewing period approaches, we wanted to kindly follow up on the responses we provided to your thoughtful reviews.
>
> Within the first week, we carefully addressed each of your questions to the best of our ability. Over the past two weeks, we have been eagerly awaiting your feedback but have not yet received any further discussion.
>
> We sincerely look forward to any additional comments or concerns you may have and will do our best to address them promptly within the remaining time.
>
> Thank you again for your consideration. We understand how busy this period can be, and your dedication to the review process means a great deal.
>
> Best regards,  \
> The Authors

---

### Author Response · Authors · 2024-11-23
**Summary of Paper Revision**

We thank all reviewers for their constructive feedback, and we have responded to each reviewer individually. We have also uploaded a **Paper Revision** including additional results and illustrations:

* $\\textrm{\\color{blue}Section 6.3 lines 483-485}$ (Page 9): Throughput improvement of our SimLayerKV. (Reviewer **KtMT**, Reviewer **YxRN**)
* $\\textrm{\\color{blue}Table 5}$ (Page 16): Pseudo code in torch style for our SimLayerKV-prefilling with flash attention. (Reviewer **KtMT**)
* $\\textrm{\\color{blue}Table 8}$ (Page 16):  Experiment results on combining SimLayerKV and intra-layer KV cache compression
method SnapKV. (Reviewer **KtMT**, Reviewer **UyjG**)
* $\\textrm{\\color{blue}Table 10}$ (Page 19):   Performance comparison of SimLayerKV and SqueezeAttention under similar compression ratio. (Reviewer **YxRN**)
* $\\textrm{\\color{blue}Table 11}$ (Page 19):   Performance on larger models (LLaMA3-70B-Instruct), and compression ratio across
different datasets in Longbench benchmark. (Reviewer **UyjG**)
* $\\textrm{\\color{blue}Table 12}$ (Page 19):   Effect of hyperparameters on lazy layer identification using LLama3-8B-Instruct. (Reviewer **KtMT**, Reviewer **YxRN**, Reviewer **fSad**)

---

### Meta-Review · Area_Chair_avdU · 2024-12-18

**Metareview:**

Summary:
The paper introduces SimLayerKV, a method for reducing KV cache memory usage in LLM inference by identifying and compressing "lazy" layers that primarily attend to initial and recent tokens. The approach requires no model training and can achieve significant memory reduction while maintaining model performance.

Main Strengths:

- Novel approach focusing on inter-layer KV cache compression
- Training-free and simple to implement
- Compatible with other compression methods like SnapKV
- Demonstrates good empirical results across multiple models and benchmarks
- Significant throughput improvements (up to 2.17x)

Main Weaknesses:

- Several hyperparameters require tuning, though additional experiments show robustness
- Some performance degradation on complex tasks like Multiple Queries NIAH
- Lack of comprehensive runtime comparisons with state-of-the-art methods initially
- Theoretical analysis of lazy layer behavior could be strengthened

**Additional Comments On Reviewer Discussion:**

Outcomes from Author-Reviewer Discussion:
The authors have addressed many initial concerns through their responses:

- Demonstrated hyperparameter robustness with additional experiments
- Added throughput comparisons showing 1.44-2.17x improvements
- Showed orthogonality with SnapKV, achieving 1.7x compression together
- Added comparisons with SqueezeAttention showing better performance
- Provided results on larger models (LLaMA-70B)

Reviewer Agreement/Disagreement:
Initial ratings ranged from 5 to 6. After author responses, two reviewers increased their scores, citing:

- Added efficiency results
- Clarified hyperparameter sensitivity
- Demonstrated compatibility with other methods
- Added comparisons with recent work

Suggestions for Improvement:

- Include more comprehensive runtime comparisons with SOTA methods
- Strengthen theoretical analysis of lazy layer behavior
- Consider dynamic adaptation for multi-round conversations
- Address performance degradation on complex tasks
- Add more discussion of hyperparameter selection process

Current consensus remains below threshold though

---

### Decision · Program_Chairs · 2025-01-22

Reject